# ADVISER-ACTOR-CRITIC: REDUCING STEADY-STATE ERROR IN REINFORCEMENT LEARNING FOR ROBOTICS CONTROL

## ABSTRACT

High-precision control tasks present substantial challenges for reinforcement learning (RL) algorithms, frequently resulting in suboptimal performance attributed to network approximation inaccuracies and inadequate sample quality. While existing RL frameworks can achieve task completion at coarse precision levels, steady-state tracking errors remain a critical limitation that prevents achieving sub-hardware-level precision. We introduce Adviser-Actor-Critic (AAC), designed to address this precision control dilemma by combining the precision of feedback control theory with the adaptive learning capability of RL and featuring an Adviser that mentors the actor to refine control actions, thereby enhancing the precision of goal attainment. Through extensive benchmark environments from gymnasium-robotics, coupled with real-world quadcopter attitude control, AAC significantly outperforms standard RL algorithms in precision-critical tasks while demonstrating an average $> 80\%$ steady-state error reduction compared to baseline methods.

## 1 INTRODUCTION

Robotic systems like dexterous hands and precision manipulators face critical control challenges in sub-millimeter accuracy tasks due to nonlinear dynamics, high-dimensional states, and parametric uncertainty sensitivity. Classical PID controllers (Li et al., 2006; Borase et al., 2021) demonstrate robustness in linear regimes but fail in nonlinear multipe-input-multiple-output (MIMO) systems owing to fixed gains and linearized models. While adaptive control (Nguyen & Dankowicz, 2015; Qu et al., 2025) and robust control (Kaloust & Qu, 1997; Zheng et al., 2025) handle bounded disturbances, their performance degrades under strong nonlinearities or unstructured uncertainties. Modern approaches include model predictive control (MPC) (Darby & Nikolaou, 2012), which employs receding-horizon optimization for constrained MIMO systems but demands precise identification and intensive computation. Geometric nonlinear methods like sliding mode control (SMC) (Utkin, 1977; Gonzalez et al., 2012) offer disturbance rejection through discontinuous control at the cost of chattering, while backstepping (Qu et al., 2024a; Swaroop et al., 2000) provides Lyapunov-stable design for strict-feedback systems but suffers from "complexity explosion" in high dimensions.

The inherent limitations of classical control paradigms, particularly in reconciling model fidelity with real-time adaptability for high-dimensional nonlinear systems, have driven interest in data-driven methodologies. This shift addresses fundamental trade-offs where rigorous model identification becomes computationally prohibitive, while purely model-free strategies often lack structural guarantees. Reinforcement learning (RL) emerges as a promising alternative, enabling direct policy learning through interaction data without requiring explicit system models. Model-free RL algorithms, such as Soft Actor-Critic (SAC) (Haarnoja et al., 2018), Proximal Policy Optimization (PPO) (Schulman et al., 2017), and Deep Deterministic Policy Gradient (DDPG) (Lillicrap et al., 2019), demonstrate superior adaptability in complex domains like contact-rich manipulation and dynamic locomotion. However, despite their adaptability, challenges like sparse rewards and function approximation errors often lead to suboptimal convergence and precision limitations, hindering industrial deployment that demands consistent performance (Pathak et al., 2017; Ramakrishnan et al., 2018; Kiran et al., 2021). These issues necessitate strategies that enhance control accuracy while preserving RL's adaptability advantages.

Conversely, model-based control frameworks offer systematic precision by leveraging learned dynamics models $\hat{f}_\theta(s_{t+1}|s_t, a_t)$ to enable receding-horizon trajectory optimization. Techniques such as Guided Policy Search (GPS) (Levine & Koltun, 2014), Temporal Difference Model Predictive Control (TD-MPC) (Hansen et al., 2024), and Probabilistic Inference for Learning Control (PILCO) (Deisenroth & Rasmussen, 2011) achieve high accuracy through probabilistic inference or deep neural dynamics modeling. However, their reliance on high-fidelity system identification often incurs significant computational costs and limits real-time applicability, as highlighted by methods like Model-Based Policy Optimization (MBPO) (Janner et al., 2019) which illustrate the inherent tension between sample efficiency and planning accuracy. This fundamental trade-off between adaptability and precision in RL for control systems has motivated the development of hybrid architectures integrating planning with policy learning (Zhuang et al., 2020). Recent advances in RL-control hybrids have begun addressing these limitations. Many of these approaches leverage RL for parameter tuning of classical controllers (e.g., autotuning PID control using Actor-Critic RL (Veldhuizen, 2022)) or for integrating safety constraints (e.g., PID Lagrangian methods for responsive safety (Stooke et al., 2020)). While others explore RL for direct policy learning (e.g., quadrotor control (Hwangbo et al., 2017)), these methods often still rely heavily on a well-designed classical control structure or a foundational understanding of system dynamics for stability and refinement. Consequently, while valuable for enhancing robustness or optimizing existing controllers, they primarily focus on safety and parameter optimization, potentially overlooking systematic error compensation mechanisms crucial for high-precision tracking tasks.

We propose an Adviser-Actor-Critic(AAC) framework that resolves this dichotomy through a novel error mitigation mechanism. Unlike conventional paradigms that directly optimize goal attainment, our approach decouples target acquisition from trajectory tracking by introducing dynamically adjusted "virtual goals" that systematically counteract error propagation. This proactive compensation strategy fundamentally differs from Hindsight Experience Replay (HER) (Andrychowicz et al., 2018), which performs retrospective goal relabeling. As demonstrated in Figure1, our framework computes offset virtual targets during execution to cancel residual errors at terminal states, thereby transforming cumulative control inaccuracies inherent in physical systems into correctable trajectory deviations. This architectural innovation enables simultaneous achievement of adaptability through policy learning and precision via error-canceling trajectory planning, addressing critical limitations in both model-free and model-based paradigms.

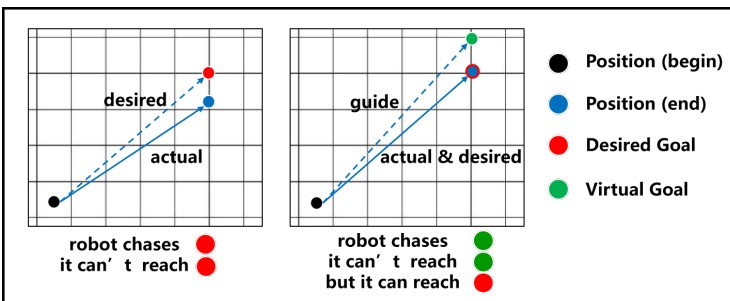

Figure 1: Conceptual example demonstrating adviser-guided precision control.

The AAC framework merges RL with control theory in a two-part system. First, an RL agent learns basic strategies through trial and error. Then, traditional controllers (like those used in engineering) adjust these strategies to improve precision in tasks requiring exact control, such as robotic movement or orientation. This method excels in applications needing high accuracy by integrating established control techniques. However, it has limitations: it works best for specific tracking tasks, requires stable environments with slow changes, and overly forceful adjustments from controllers can disrupt learning by causing mismatches between the agent's actions and the desired behavior, leading to unstable training.

We address DRL's limitations in high-precision robotic control through three key contributions. First, we propose a **Hybrid Dual-Loop Control** framework that integrates classical feedback principles with DRL, combining goal-conditioned policies and complementary compensation strategies to systematically reduce steady-state errors. Second, we develop a **Lightweight Adviser Module**, a proportional-integral controller enabling generalizable error correction across Cartesian and SO(3)

spaces with only 2 tunable parameters. Third, we conduct **Benchmark Validation** demonstrating ¿80% steady-state error reduction in 6 gymnasium-robotics environments and physical quadcopter experiments.

## 2 PROBLEM MODELING

We begin by formalizing the control problem through a nonlinear time-invariant system governed by:

$$
\begin{cases}
\dot{\boldsymbol{s}} = \boldsymbol{f}(\boldsymbol{s}, \boldsymbol{a}), & \text{(System dynamics)} \\
\boldsymbol{g}_a = \boldsymbol{g}_a(\boldsymbol{s}), & \text{(Achieved goal)} \\
\boldsymbol{g}_d = \boldsymbol{g}_d(t), & \text{(Desired goal)} \\
\boldsymbol{a} = \boldsymbol{\pi}(\boldsymbol{s}, \boldsymbol{g}_a, \boldsymbol{g}_d), & \text{(Control policy)}
\end{cases}
\tag{1}
$$

here, $t \in \mathbb{R}^+$ is time, $\boldsymbol{s} \in \mathbb{R}^p$ denotes the state vector, $\boldsymbol{a} \in \mathbb{R}^m$ represents actions, and $\boldsymbol{g}_a, \boldsymbol{g}_d \in \mathbb{R}^n$ correspond to achieved and desired goals respectively. The dynamics function $\boldsymbol{f} : \mathbb{R}^p \times \mathbb{R}^m \to \mathbb{R}^p$ captures the system's nonlinear behavior. The achieved goal $\boldsymbol{g}_a$ depends solely on the current state $\boldsymbol{s}$, while the desired goal $\boldsymbol{g}_d$ is specified as a function of time $t$. The desired goal trajectory $\boldsymbol{g}_d(t)$ may be randomly generated or represent actual command signals.

To enable effective policy learning, we define an extended observation space, $\boldsymbol{s}_e$, which serves as a convenient representation for the input to our control policy $\boldsymbol{\pi}$:

$$
\boldsymbol{s}_e = [\boldsymbol{s}, \boldsymbol{g}_a, \boldsymbol{g}_d] \in \mathbb{R}^{2n+p}
\tag{2}
$$

This extended observation $\boldsymbol{s}_e$ explicitly comprises the current system state $\boldsymbol{s}$, the achieved system output $\boldsymbol{g}_a$, and the desired system output $\boldsymbol{g}_d$. The tracking error $\boldsymbol{e}$ is then computed as:

$$
\boldsymbol{e} = \boldsymbol{g}_a - \boldsymbol{g}_d
\tag{3}
$$

The control objective is to derive a policy $\boldsymbol{\pi} : \mathbb{R}^p \times \mathbb{R}^n \times \mathbb{R}^n \to \mathbb{R}^m$ that maps these inputs $(\boldsymbol{s}, \boldsymbol{g}_a, \boldsymbol{g}_d)$ to the control inputs $\boldsymbol{a}$, aiming to maximize the cumulative reward:

$$
R = \sum_{t=0}^{\infty} r(\boldsymbol{g}_a, \boldsymbol{g}_d, \boldsymbol{a})
\tag{4}
$$

The reward function $r(\cdot)$ is designed to balance two critical aspects: penalizing deviations between $\boldsymbol{g}_a$ and $\boldsymbol{g}_d$ (tracking accuracy) and accounting for action $\boldsymbol{a}$ costs (control efficiency).

## 3 RELATED WORKS

Modern reinforcement learning (RL) systems for high-precision robotic control converge on four complementary methodologies: **Reward Shaping** for error-sensitive policy gradients, **Hindsight Experience Replay (HER)** for sparse-reward mitigation, **Integrator Feedback** for steady-state error mitigating, and **Model Predictive Path Integral (MPPI)** for real-time control sequence optimization.

**Reward shaping** modifies the reward function by incorporating *domain-specific shaping potentials* (e.g., distance-to-goal penalties) to amplify gradient signals for policy optimization (Burda et al., 2018; Hu et al., 2020). While effective for accelerating convergence in robotic manipulation tasks, improper shaping can distort the reward landscape, leading to suboptimal policies (Hu et al., 2020). This limitation motivates adaptive shaping mechanisms that balance task-specific guidance with reward function fidelity.

**Hindsight Experience Replay (HER)** enhances sample efficiency in goal-conditioned RL by relabeling failed trajectories with achieved goals, effectively converting failures into pseudo-successes (Andrychowicz et al., 2018). This "failure-as-curriculum" paradigm is particularly impactful in sparse-reward scenarios (e.g., precision grasping), though it risks overfitting to suboptimal subgoals when reward horizons are misaligned (Moro et al., 2022). Recent extensions like *Time-Aggregated HER (TA-HER)* mitigate this by enforcing temporal consistency in goal relabeling.

**Integrator feedback** augments observation spaces with cumulative error states (e.g., $\int e(t)dt$), enabling policies to implicitly compensate for steady-state errors akin to PID control (Tracey et al.,

2023). However, this approach introduces challenges: (1) *integral windup* in high-noise environments, and (2) increased observation dimensionality that degrades generalization in dynamic tasks. These trade-offs necessitate hybrid designs that selectively activate integrator terms based on error thresholds.

**Model Predictive Path Integral (MPPI)** is a sampling-based control method that optimizes action sequences through stochastic trajectory rollouts, leveraging gradient-free optimization for non-convex cost functions (Williams et al., 2015). Recent work combines MPPI with RL (e.g., *RL-driven MPPI* (Qu et al., 2024b)) to adaptively refine cost functions, bridging the gap between open-loop planning and closed-loop policy learning.

## 4 ADVISER-ACTOR-CRITIC

### 4.1 FRAMEWORK

Figure 2: **Interact with Environment:** Policy operates through transformed operator $\pi_\phi^{adv}(s, g_a, g_d) = \pi_\phi(s, g_a, \text{adv}(g_a, g_d))$, where the adviser generates virtual goal $g_v$ to condition Actor's decisions, integrating environmental observation $s$, achieved goal $g_a$, and virtual goals $g_v$; **Learn from Experience:** Actor-Critic framework with policy network $\pi_\phi(\cdot|s_e)$ generating actions and value network $Q_\theta(s_e, a)$ estimating returns.

The AAC framework, shown in Figure 2, employs a decoupled architecture with two specialized components: an actor-training module (right) and an adviser-enhanced deployment system (left), enabling plug-and-play integration of guidance mechanisms without modifying core policies. This modular design allows reinforcement learning actors to optionally leverage task-specific advisers during exploration, such as physics constraints for robotic manipulation or traffic rules for autonomous driving, ensuring precision-critical applications maintain interpretability while improving adaptability.

This decoupled design underpins AAC's central contribution: the "virtual goal" strategy, which systematically modifies target states to amplify trajectory errors during training. By coupling control-theoretic principles with RL optimization, this mechanism expedites exploration convergence through exaggerated error signals while ensuring stability. Although the adviser breaks the MDP assumption by using history to shape goals, the actor still works well because it treats the virtual goal like any random target it was trained on—effectively seeing the world as Markovian. See Appendix A for details.

### 4.2 ADVISER IMPLEMENTATION

The adviser module represents the conceptual centerpiece of our framework, offering systematic error compensation through dynamic goal modulation. **The theoretical validity and convergence**

**guarantees of this control strategy are rigorously established in Appendix B.** Grounded in control-theoretic formalism, we establish its operational paradigm through the following mathematical formulation:

**Definition 4.1** (Adviser). The adviser module operates as a temporal compensator with memory persistence, synthesizing corrective virtual goals ($\boldsymbol{g}_v$) through spatiotemporal analysis of goal trajectories. Formally, given sequences of achieved goals $\{\boldsymbol{g}_a^{(t-k)}\}_{k=0}^{K}$ and desired goals $\{\boldsymbol{g}_d^{(t-k)}\}_{k=0}^{K}$, where $K$ denotes the number of past time steps considered for historical analysis, the advisory transformation generates regulated targets via:

$$\boldsymbol{g}_v^{(t)} = \mathrm{adv}\left(\boldsymbol{g}_a^{(t)}, \boldsymbol{g}_d^{(t)}; \mathcal{M}_t\right) \tag{5}$$

where $\mathcal{M}_t$ denotes the time-variant memory construct encapsulating historical interaction data $\{(\boldsymbol{g}_a^{(t-k)}, \boldsymbol{g}_d^{(t-k)})\}_{k=1}^{K}$. This memory architecture enables $\boldsymbol{g}_v^{(t)}$ synthesis by concurrently analyzing long-term trends and dynamically adapting to immediate contextual deviations.

The adviser is conceptualized as a versatile component, capable of leveraging a wide array of advanced control methodologies. However, for this study, we deliberately adopt a more straightforward approach: a Proportional-Integral (PI) controller. This choice is primarily driven by the desire to facilitate both theoretical analysis and reader comprehension. To ensure stable and effective control, our implementation utilizes a modified PI controller (Li et al., 2006). This controller is specifically designed to prevent integral windup—a common issue where large sustained errors cause the integral term to accumulate excessively, leading to significant overshoots and slow recovery—and to keep the control output within practical limits. The intelligent management of its integral component is detailed in Equation 6.

$$\boldsymbol{g}_v(t) = K_p \boldsymbol{e}(t) + K_i \boldsymbol{I}(t)$$

$$\boldsymbol{I}(t) = \begin{cases} \boldsymbol{\sigma}, & \text{if } \boldsymbol{I}(t-\Delta t) + \alpha \boldsymbol{e}(t)\Delta t > \boldsymbol{\sigma} \\ -\boldsymbol{\sigma}, & \text{if } \boldsymbol{I}(t-\Delta t) + \alpha \boldsymbol{e}(t)\Delta t < -\boldsymbol{\sigma} \\ \boldsymbol{I}(t-\Delta t) + \alpha \boldsymbol{e}(t)\Delta t, & \text{otherwise} \end{cases} \tag{6}$$

Here, $\boldsymbol{e}(t) = \boldsymbol{g}_a(t) - \boldsymbol{g}_d(t)$ is the error, and $I(t)$ is the accumulated error. $K_p$ is the proportional gain, which scales the immediate impact of the current error $\boldsymbol{e}(t)$ on the control output. $K_i$ is the integral gain, which determines the influence of the accumulated error $\boldsymbol{I}(t)$. The parameter $\sigma$ defines the saturation limits for the integral term $\boldsymbol{I}(t)$, ensuring it remains within $[-\sigma, \sigma]$ and thereby directly implementing an anti-windup mechanism. The parameter $\alpha \in \{0, 1\}$ acts as a switch: when $\alpha = 1$, the integral term $\boldsymbol{I}(t)$ accumulates the error $\boldsymbol{e}(t)$, but only when the error magnitude $|e(t)|$ is within a specific range defined by $\sigma$. This prevents the integral term from growing uncontrollably when the error is very large. $\Delta t$ denotes the sampling interval (or control update period), assumed to be constant across time steps. This design achieves three main goals: (1) It prevents the integral term from 'winding up' excessively, (2) It helps ensure the control output stays within practical boundaries by limiting the integral contribution, and (3) It helps maintain high accuracy once the system has settled. As detailed in Algorithm 1, this approach is well-suited for reinforcement learning because it naturally keeps control actions within safe limits and provides strong guarantees for stable system behavior by adapting how the integral term accumulates error.

> **Degenerate Configuration Analysis:** Under parameter configuration $K_p = 1.0, K_i = 0.0$ the virtual goal generation reduces to:
>
> $$\boldsymbol{g}_v = \boldsymbol{g}_a - \boldsymbol{e} = \boldsymbol{g}_a - (\boldsymbol{g}_a - \boldsymbol{g}_d) = \boldsymbol{g}_d$$
>
> This degenerate form reduces error compensation mechanisms, rendering the Adviser module functionally identical to the baseline Actor policy. The framework thereby ensures performance **at minimum comparable to** conventional actor-critic implementations.

## 5 EXPERIMENTS

Our evaluation of AAC is structured in three stages. First, we present a minimal illustrative example to elucidate the adviser's core mechanism. Subsequently, benchmark tests (Figure 3) are conducted to

---

**Algorithm 1** Dual-Mode Adviser Implementation // $K_p = 1.0$ yields near optimal performance

**Position Advisory Controller:**

**Require:**

1: $\mathbf{p}_a, \mathbf{p}_d \in \mathbb{R}^3$ // Actual and desired position vectors (m)

2: $K_p, K_i, \sigma, \Delta t$ // $\sigma$: Position error threshold (m)

**Ensure:**

3: $\tilde{\mathbf{p}}$ // Compensated position output

4: $\Delta \mathbf{p} \leftarrow \mathbf{p}_a - \mathbf{p}_d$ // Compute position error vector

5: **if** $\|\Delta \mathbf{p}\| \leq \sigma$ **then**

6:    $\mathbf{I} \leftarrow \text{clip}(\mathbf{I} + \Delta \mathbf{p} \cdot \Delta t, -\sigma, \sigma)$ // Integral state clamping within bounds

7: **end if**

8: $\tilde{\mathbf{p}} \leftarrow \mathbf{p}_a - (K_p \Delta \mathbf{p} + K_i \mathbf{I})$ // PI regulation in Cartesian coordinates

**Orientation Advisory Controller:**

**Require:**

1: $\mathbf{q}_a, \mathbf{q}_d \in \mathbb{H}$ // Observed and reference quaternions

2: $K_p, K_i, \sigma, \Delta t$ // $\sigma$: Tolerance threshold (rad)

**Ensure:**

3: $\tilde{\mathbf{q}}$ // Compensated orientation quaternion

4: $\mathbf{q}_{err} \leftarrow \mathbf{q}_a \otimes \mathbf{q}_d^{-1}$ // Quaternion error calculation ($\approx [\cos \frac{\theta}{2}, \sin \frac{\theta}{2} \mathbf{n}]$)

5: $\mathbf{v} \leftarrow \text{Im}(\mathbf{q}_{err})$ // Extract rotation vector component

6: **if** $\|\mathbf{v}\| \leq \sigma/2$ **then** // Check against angular bound

7:    $\mathbf{I} \leftarrow \text{clip}(\mathbf{I} + \mathbf{v} \cdot \Delta t, -\sigma/2, \sigma/2)$ // Angular integral state bounding

8: **end if**

9: $\Delta \mathbf{q} \leftarrow [\sqrt{1 - \|\mathbf{v}\|^2}, -K_p \mathbf{v} - K_i \mathbf{I}]$ // Construct corrective quaternion

10: $\tilde{\mathbf{q}} \leftarrow \Delta \mathbf{q} \otimes \mathbf{q}_a$ // Apply orientation correction through composition

---

compare its performance against state-of-the-art Goal-Conditioned Reinforcement Learning (GCRL) methods under standardized conditions. Finally, real-world experiments validate its effectiveness on a quadrotor platform subjected to environmental disturbances (Figure 4). All simulations were performed on a testbed configured with Windows 10, an Intel Core i7-12700K processor, 64 GB RAM, and an NVIDIA GeForce RTX 3060 Ti GPU. Comprehensive experimental settings and algorithm configurations are detailed in Appendix C.

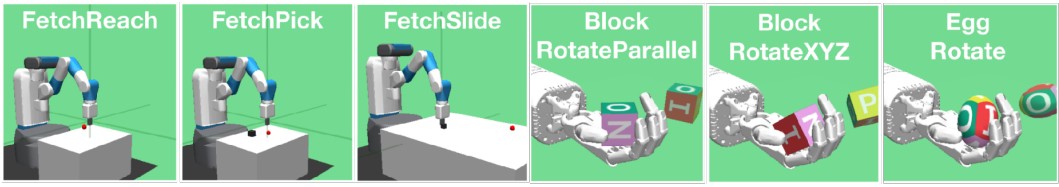

Figure 3: Benchmark environments from `gymnasium-robotics` (Plappert et al., 2018): (a) **Fetch Manipulator** - 7-DOF serial manipulator with 4D continuous action space; (b) **Shadow Hand** - 24-DOF dexterous hand with 20D torque control.

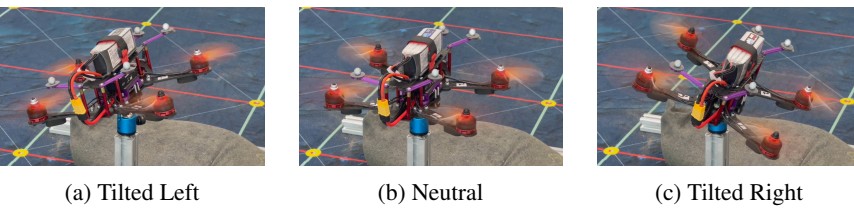

(a) Tilted Left      (b) Neutral      (c) Tilted Right

Figure 4: Illustration of the quadcopter's orientations.

## 5.1 HOW DOES THE ADVISER REDUCE STEADY-STATE ERRORS?

For the purpose of elucidating the functional mechanism of the adviser module, and not as a comprehensive performance evaluation, a canonical mass-spring-damper system is implemented. This simplified setup is governed by the second-order differential equation $m\ddot{x} + c\dot{x} + kx = F_{\text{control}}$, where the actor employs linear feedback control $F_{\text{control}} = -15.0 \cdot x - 3.0 \cdot \dot{x}$. With system parameters fixed at $m = 1.0 \, \text{kg}$, $c = 0.5 \, \text{N} \cdot \text{s/m}$, $k = 2.0 \, \text{N/m}$, and initial conditions $x_0 = 0.0 \, \text{m}$, the control objective is set to $x_d = 1.0 \, \text{m}$. This experimental configuration serves to systematically demonstrate the framework's capability to compensate for residual tracking errors through adviser-guided policy refinement (Algorithm 1).

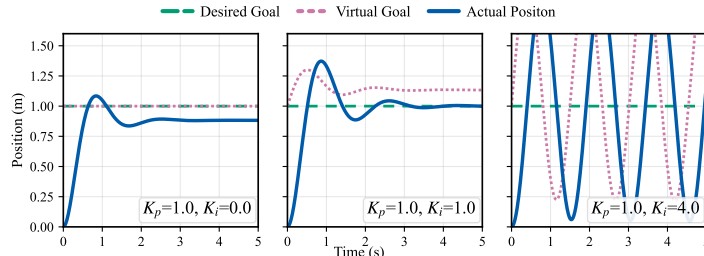

Figure 5: Adviser-Guided Policy Refinement: (left) Baseline steady-state error; (middle) Good adviser enables convergence via virtual goal adaptation; (right) Bad adviser induces instability. ($\sigma = \infty$ isolates $K_i$ effect; see Appendix F for other $\sigma$ values).

Figure. 5 illustrates the error compensation mechanism: Without the adviser (left), the system shows 10% steady-state error (0.9 m vs. 1.0 m target) due to policy suboptimality. With proper adviser integration (middle), dynamic virtual goals overcompensate the reference trajectory, enabling asymptotic convergence to $x_d$. Improper parameterization (right) causes divergent oscillations, highlighting the need for stability constraints.

## 5.2 DOES THE ADVISER IMPROVE PERFORMANCE?

We benchmark on two gymnasium-robotics environments: Fetch Manipulator and the challenging 24-DOF Shadow Hand. The Shadow Hand's complex, tendon-driven dynamics pose significant challenges: conventional methods (e.g., PID(Li et al., 2006)) fail due to linearity assumptions, while mainstream DRL (e.g., SAC(He et al., 2020), PPO(Schulman et al., 2017)) struggles with sparse rewards and approximation errors in multi-contact scenarios. Performance is assessed over 1,000 test episodes per environment, primarily using steady-state error for control precision. Other comparative results are detailed in Appendix D and Appendix E.

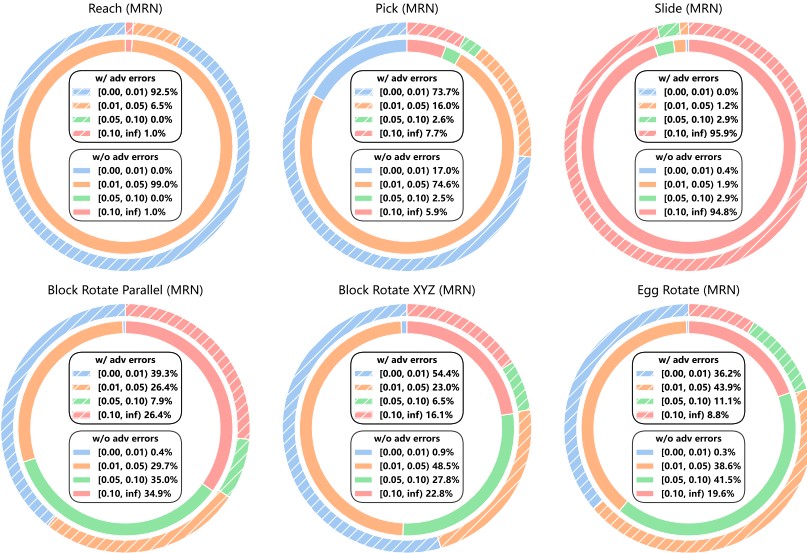

Figure 6: Steady-State Error Analysis Across 6 Benchmark Environments: Column Visualization of AAC's Control Precision. AAC is evaluated using Metric Residual Networks (MRN) (Liu et al., 2023). The performance is assessed through steady-state error across six benchmark environments. In six of these environments, AAC demonstrates satisfactory results.

Empirical analysis across six Gymnasium-Robotics environments (Figure 6) demonstrates AAC's remarkable ability to reduce steady-state error (SSE) and enable high-precision scenarios (SSE < 0.01) significantly more often than baselines. Quantitative results (Table 1) show an average 80% SSE reduction in tasks like Reach, Pick, BlockRotate Parallel/XYZ, and EggRotate.

Overall, these findings highlight the exceptional effectiveness of the AAC framework in minimizing control steady-state errors, underscoring its potential for enhancing precision in robotic control tasks.

Table 1: Performance Comparison: AAC Framework vs. Reinforcement Learning Baselines (SAC(He et al., 2020), PPO(Schulman et al., 2017), DreamerV3(Hafner et al., 2025), MRN(Liu et al., 2023), WN(Pitis et al., 2019))

| Environment | Method | SSE | | Success Rate (%) | |
|---|---|---|---|---|---|
| | | w/o Adviser | w/ Adviser | w/o Adviser | w/ Adviser |
| Reach | SAC | $0.023 \pm 0.003$ | $\mathbf{0.000 \pm 0.000}$ | $100.0 \pm 0.0$ | $100.0 \pm 0.0$ |
| | PPO | $0.027 \pm 0.004$ | $\mathbf{0.001 \pm 0.002}$ | $100.0 \pm 0.0$ | $100.0 \pm 0.0$ |
| | DreamerV3 | $0.007 \pm 0.001$ | $\mathbf{0.000 \pm 0.000}$ | $100.0 \pm 0.0$ | $100.0 \pm 0.0$ |
| | MRN | $0.036 \pm 0.003$ | $\mathbf{0.000 \pm 0.001}$ | $99.0 \pm 0.4$ | $99.0 \pm 0.4$ |
| | WN | $0.033 \pm 0.002$ | $\mathbf{0.000 \pm 0.000}$ | $100.0 \pm 0.0$ | $100.0 \pm 0.0$ |
| Pick | SAC | $0.023 \pm 0.008$ | $\mathbf{0.003 \pm 0.004}$ | $92.0 \pm 1.2$ | $\mathbf{98.0 \pm 0.6}$ |
| | PPO | $0.037 \pm 0.036$ | $\mathbf{0.005 \pm 0.005}$ | $89.3 \pm 1.4$ | $\mathbf{96.9 \pm 0.8}$ |
| | DreamerV3 | $0.019 \pm 0.015$ | $\mathbf{0.004 \pm 0.004}$ | $89.5 \pm 1.3$ | $\mathbf{98.3 \pm 0.6}$ |
| | MRN | $0.020 \pm 0.010$ | $\mathbf{0.011 \pm 0.017}$ | $\mathbf{91.7 \pm 1.2}$ | $89.6 \pm 1.4$ |
| | WN | $0.045 \pm 0.047$ | $\mathbf{0.008 \pm 0.008}$ | $83.5 \pm 1.5$ | $\mathbf{93.9 \pm 1.0}$ |
| Slide | SAC | $\mathbf{0.551 \pm 0.246}$ | $0.583 \pm 0.233$ | $\mathbf{2.1 \pm 0.7}$ | $1.7 \pm 0.6$ |
| | PPO | $\mathbf{0.535 \pm 0.234}$ | $0.543 \pm 0.226$ | $\mathbf{3.4 \pm 0.8}$ | $1.5 \pm 0.6$ |
| | DreamerV3 | $\mathbf{0.549 \pm 0.235}$ | $0.563 \pm 0.230$ | $\mathbf{2.0 \pm 0.6}$ | $1.7 \pm 0.6$ |
| | MRN | $\mathbf{0.563 \pm 0.227}$ | $0.567 \pm 0.222$ | $\mathbf{2.3 \pm 0.7}$ | $1.2 \pm 0.5$ |
| | WN | $\mathbf{0.545 \pm 0.239}$ | $0.561 \pm 0.227$ | $\mathbf{2.9 \pm 0.8}$ | $1.4 \pm 0.5$ |
| Block Rotate Parallel | SAC | $0.064 \pm 0.048$ | $\mathbf{0.017 \pm 0.030}$ | $80.9 \pm 1.7$ | $\mathbf{90.1 \pm 1.3}$ |
| | PPO | $0.053 \pm 0.026$ | $\mathbf{0.011 \pm 0.018}$ | $88.7 \pm 1.4$ | $\mathbf{94.9 \pm 1.0}$ |
| | DreamerV3 | $0.051 \pm 0.017$ | $\mathbf{0.005 \pm 0.007}$ | $97.1 \pm 0.7$ | $\mathbf{99.4 \pm 0.3}$ |
| | MRN | $0.268 \pm 0.471$ | $\mathbf{0.239 \pm 0.496}$ | $65.2 \pm 2.1$ | $\mathbf{73.6 \pm 2.0}$ |
| | WN | $0.043 \pm 0.020$ | $\mathbf{0.004 \pm 0.005}$ | $95.2 \pm 1.0$ | $\mathbf{97.2 \pm 0.8}$ |
| Block Rotate XYZ | SAC | $0.092 \pm 0.083$ | $\mathbf{0.057 \pm 0.083}$ | $72.3 \pm 2.0$ | $\mathbf{82.0 \pm 1.7}$ |
| | PPO | $0.067 \pm 0.046$ | $\mathbf{0.015 \pm 0.023}$ | $80.5 \pm 1.8$ | $\mathbf{91.6 \pm 1.2}$ |
| | DreamerV3 | $0.048 \pm 0.026$ | $\mathbf{0.006 \pm 0.011}$ | $89.4 \pm 1.3$ | $\mathbf{95.6 \pm 0.9}$ |
| | MRN | $0.074 \pm 0.060$ | $\mathbf{0.057 \pm 0.107}$ | $77.2 \pm 1.8$ | $\mathbf{83.8 \pm 1.6}$ |
| | WN | $0.052 \pm 0.029$ | $\mathbf{0.010 \pm 0.016}$ | $88.0 \pm 1.4$ | $\mathbf{95.2 \pm 1.0}$ |
| Egg Rotate | SAC | $0.062 \pm 0.025$ | $\mathbf{0.013 \pm 0.018}$ | $87.2 \pm 1.5$ | $\mathbf{95.5 \pm 1.0}$ |
| | PPO | $0.062 \pm 0.046$ | $\mathbf{0.025 \pm 0.033}$ | $83.6 \pm 1.6$ | $\mathbf{89.6 \pm 1.4}$ |
| | DreamerV3 | $0.050 \pm 0.021$ | $\mathbf{0.020 \pm 0.017}$ | $92.9 \pm 1.2$ | $\mathbf{96.9 \pm 0.8}$ |
| | MRN | $0.072 \pm 0.042$ | $\mathbf{0.028 \pm 0.029}$ | $80.4 \pm 1.8$ | $\mathbf{91.2 \pm 1.3}$ |
| | WN | $0.052 \pm 0.027$ | $\mathbf{0.015 \pm 0.017}$ | $88.7 \pm 1.4$ | $\mathbf{97.6 \pm 0.7}$ |

Note: (1) Values are mean $\pm$ 2 std (bootstrap: 1000 resamples of 500 episodes from 1000 runs). (2) Success criteria and SSE definitions: SSE < 0.05 (first 3 environments, meter-based) and < 0.10 (last 3 environments, radius-based). (3) Bold values indicate better performance.

However, AAC exhibits critical limitations in environments like `Slide`, where the absence of a steady-state regime makes traditional SSE metrics incompatible. This highlights AAC's current algorithmic constraints in handling tasks requiring prolonged precision or stochastic contact dynamics. While validating AAC's superior efficacy in explicit steady-state tasks, these findings also underscore the need for further refinement to address challenges in complex, dynamic environments.

## 5.3 DEPLOYMENT ON A REAL-WORLD QUADCOPTER

To evaluate the AAC framework's effectiveness in practical scenarios, we conducted physical experiments testing its ability to refine suboptimal actors under real-world noisy conditions. Specifically, we implemented adviser-guided control for quadcopter attitude regulation tasks with various non-ideal factors and characterized transient responses. Experimental results quantified significant improvements in steady-state angular errors and settling time, demonstrating the framework's capability to maintain sub-degree attitude precision in practical quadcopter operation.

Experimental validation on a 1.40kg Pixhawk4/ROS quadcopter platform confirmed the efficacy of the Adviser-Actor-Critic (AAC) framework. Simulation-trained controllers maintained attitude stability on hardware (1° steady-state error due to sim-to-real shift), and the Adviser module (Algorithm 1) further reduced angular error to $0.03°$ by compensating for unmodeled perturbations. However, this enhanced precision introduced a performance tradeoff in transient response (Table 2): AAC's 3.25 s settling time was slower than the native Pixhawk4 cascade PID's 0.91 s. This disparity is attributed to the Adviser's outer-loop integral action causing phase lag, contrasting with PID's direct inner-loop error correction.

Table 2: Optimized Adviser Parameters for Attitude Control (5 mm CoM Offset)

| $K_i$ | $\sigma$ | SSE (°) | Settling Time (s) |
|-------|----------|---------|-------------------|
| 0.00 | 0.0 | 1.05 | $\infty$ |
| 0.05 | 2.0 | 0.02 | 3.85 |
| 0.10 | 2.0 | 0.03 | 3.25 |
| 0.20 | 2.0 | 0.02 | 4.21 |
| 0.05 | 10.0 | 0.03 | 3.72 |
| 0.10 | 10.0 | 0.03 | 3.55 |
| 0.20 | 10.0 | 0.03 | 4.78 |
| Cascade | PID | 0.03 | 0.91 |

Note: (1) 5° step command inputs; (2) Fixed proportional gain $K_p = 1.0$.

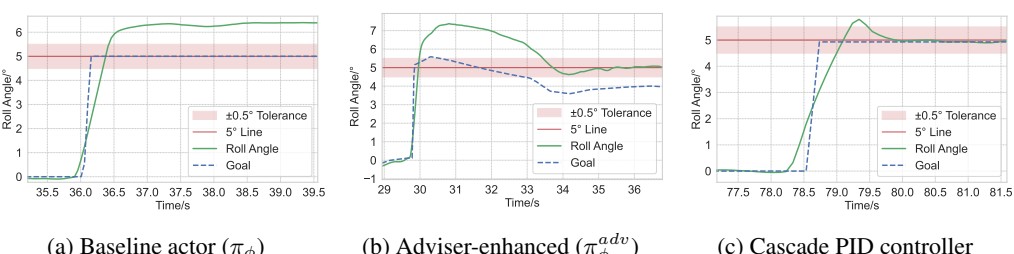

(a) Baseline actor ($\pi_\phi$)  (b) Adviser-enhanced ($\pi_\phi^{adv}$)  (c) Cascade PID controller

Figure 7: Comparative performance analysis of attitude control strategies. Attitude control performance comparison. AAC achieves sub-degree precision via sequential compensation (Algorithm 1) in hardware validation.

# 6 DISCUSSION AND CONCLUSION

In conclusion, the Adviser-Actor-Critic (AAC) framework successfully bridges classical feedback control principles with modern reinforcement learning, delivering significant enhancements in control precision for complex robotic systems. Our experimental results demonstrate that AAC achieves an average $> 80\%$ steady-state error reduction across 6 gymnasium-robotics benchmarks (Fetch/ShadowHand) and attains sub-degree precision in real-world quadcopter attitude control, consistently outperforming standard RL algorithms. Theoretically, we have rigorously established that AAC guarantees asymptotic stability and reliable steady-state error elimination under qualified actor conditions. The modular architecture of AAC facilitates plug-and-play integration, offering both improved adaptability and enhanced interpretability for robotic control applications.

**Limitations.** While the AAC framework demonstrates significant performance improvements, its effectiveness is contingent upon several key constraints: (1) The current implementation is primarily applicable to quasi-static processes where reference trajectories evolve at slower rates than the underlying system dynamics; (2) The framework requires actor policies with basic decoupling capabilities to ensure stable operation; (3) Performance is dependent on the quality of the baseline actor policy, with severely suboptimal actors presenting challenges for effective adviser guidance.

**Future works.** Future research directions will focus on extending the AAC framework to highly dynamic environments, developing adaptive mechanisms for automated adviser parameter tuning, and enhancing robustness when working with severely suboptimal baseline actors.

**Reproducibility statement.** The implementation code is available at: `https://anonymous.4open.science/r/Adviser-Actor-Critic-8AC5/`. Comprehensive experimental settings and configuration details are documented in Appendix C.

**Large Language Models.** We thank Large Language Models for their assistance in code development, algorithm implementation, and technical writing refinement throughout this research.

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

# A  SCENARIO DEFINITIONS AND DEPLOYMENT ANALYSIS

Based on the problem modeling presented in Section 2, we define and contrast two scenarios: the "Random Target Scenario" (an MDP) where the desired goal $g_d$ is randomly generated, and the "Virtual Target Scenario" (a POMDP) where $g_d$ is influenced by an unobservable internal state of an "Adviser". The key differences between these two scenarios are summarized in Table 3.

Table 3: Scenario Comparison

| Characteristic | Random Target Scenario | Virtual Target Scenario |
|---|---|---|
| **True State** | $s_{e,t} = [s_t, g_{a,t}, g_{d,t}]$ | $x_t = [s_t, g_{a,t}, g_{d,t}, I_t]$ |
| **Agent's Observation** | $o_t = [s_t, g_{a,t}, g_{d,t}]$ | $o_t = [s_t, g_{a,t}, g_{v,t}]$ |
| **Desired Goal $g_{d,t}$ / Virtual Goal $g_{v,t}$ Source** | Exogenous, randomly generated. | Generated by Adviser with unobservable internal state $I_t$. |

We now analyze the technical feasibility of deploying an Actor $\pi_{\text{rand}}$ trained in the random target scenario directly into the virtual target scenario. This deployment is fundamentally feasible due to three critical aspects.

Firstly, input interface compatibility is ensured. The policy $\pi_{\text{rand}}$ is defined as a mapping $\pi_{\text{rand}} : \mathcal{S}_e \to \mathcal{P}(\mathcal{A})$, where $\mathcal{S}_e = \mathbb{R}^{p+2n}$ is the extended observation space. In the random target scenario, $\pi_{\text{rand}}$ expects an observation $o_t = [s_t, g_{a,t}, g_{d,t}]$. In the virtual target scenario, the agent receives an observation $o_t = [s_t, g_{a,t}, g_{v,t}]$. This direct correspondence, where the virtual goal $g_{v,t}$ takes the place of the desired goal $g_{d,t}$ in the observation vector, guarantees that the domain of $\pi_{\text{rand}}$ perfectly aligns with the observable output of the POMDP environment, allowing for direct execution: $a_t \sim \pi_{\text{rand}}(\cdot|o_t)$. This implies that $g_{v,t}$ must be structurally and semantically compatible with $g_{d,t}$.

Secondly, the reactive nature of $\pi_{\text{rand}}$ facilitates its seamless integration. As an MDP-trained policy, $\pi_{\text{rand}}$ is inherently memoryless, determining actions $a_t$ solely based on the current observation $o_t$. It does not require maintaining a belief state $b_t(I)$ over the unobservable internal state $I_t$ of the Adviser, nor does it rely on the history of observations $h_t$. From the Actor's perspective, the observed virtual goal $g_{v,t}$ is simply an exogenous, fully revealed component of the current observation $o_t$, irrespective of its generation mechanism. This reactive operational mode ensures the Actor can continuously generate control actions within the virtual target environment, maintaining functional viability.

Thirdly, the Virtual Target Scenario can be understood as a specific instance of the Random Target Scenario. The Random Target Scenario defines a broad context where the desired goal $g_{d,t}$ is generated exogenously and randomly, implying it can take on any value within the defined goal space $\mathcal{G}$ according to some distribution. In contrast, the Virtual Target Scenario features a more specific, albeit complex, generative process for the observed virtual goal $g_{v,t}$ driven by an Adviser with an unobservable internal state $I_t$. Crucially, the virtual goals $g_{v,t}$ produced by this Adviser are assumed to fall within the same domain and range of values as the desired goals $g_{d,t}$ encountered in the Random Target Scenario. Therefore, from the Actor's perspective, the Virtual Target Scenario presents a particular sequence of virtual goals that are a subset of, or at least consistent with, the goal distributions it was trained to handle in the more general Random Target Scenario.

# B  ASSUMPTIONS AND PROOFS FOR STEADY-STATE ERROR ELIMINATION

This appendix rigorously establishes the conditions under which the Adviser-Actor-Critic (AAC) framework, particularly the PI-based adviser, guarantees the elimination of steady-state errors. The analysis relies on standard control-theoretic assumptions regarding system behavior and the properties of the learned actor policy, leveraging a local linearization approach to enable time-domain and frequency-domain perspectives.

**System Model and Error Definition**    As defined in Section 2, the system dynamics are given by $\dot{\boldsymbol{s}} = \boldsymbol{f}(\boldsymbol{s}, \boldsymbol{a})$, with an output mapping $\boldsymbol{g}_a = \boldsymbol{\phi}(\boldsymbol{s})$. The desired output is $\boldsymbol{g}_d = \boldsymbol{\psi}(t)$. The tracking error, consistent with its definition in Section 2, is $\boldsymbol{e}(t) = \boldsymbol{g}_a(t) - \boldsymbol{g}_d(t)$.

**Adviser and Actor Interaction**    The Adviser module generates a virtual goal $\boldsymbol{g}_v(t)$ based on the current tracking error $\boldsymbol{e}(t)$ using a modified Proportional-Integral (PI) control law, as detailed in Algorithm 1 and Equation 6 of the main text:

$$\boldsymbol{g}_v(t) = \boldsymbol{g}_d(t) - \left[ K_p \boldsymbol{e}(t) + K_i \int_0^t \boldsymbol{e}(\tau) d\tau \right] \tag{7}$$

The Actor's policy $\boldsymbol{\pi}$ takes the current system state $\boldsymbol{s}(t), \boldsymbol{g}_a(t)$, and the virtual goal $\boldsymbol{g}_v(t)$ as input to produce the control action $\boldsymbol{a}(t)$, as introduced in Section 2 (specifically, in the definition of the control policy $\boldsymbol{\pi}$):

$$\boldsymbol{a}(t) = \boldsymbol{\pi}(\boldsymbol{s}(t), \boldsymbol{g}_a(t), \boldsymbol{g}_v(t)) \tag{8}$$

To facilitate the analysis of the closed-loop system using linear control theory, we make the following assumptions regarding the system dynamics and the learned actor policy. Assumptions B.1 and B.2 characterize the linearized dynamics of the plant, which, when coupled with the actor's policy, form the inner control loop.

**Assumption B.1** (Local Linearity of Error Dynamics). In a sufficiently small neighborhood around the desired operating point (where $\boldsymbol{g}_a = \boldsymbol{g}_d$), the system dynamics relating the control action $\boldsymbol{a}(t)$ to the tracking error $\boldsymbol{e}(t)$ can be approximated by a linear time-invariant (LTI) model. Specifically, the $k$-th order derivative of the $i$-th error component $e_i^{(k)}$ is modeled as a linear combination of lower-order derivatives $e_j^{(l)}$ (for $l < k$) and the control input deviation variables $u_j$, plus a disturbance term $\tilde{d}_i$.

$$e_i^{(k)} = -\sum_{j=1}^{n} \sum_{l=0}^{k-1} a_{ijl} e_j^{(l)} + \sum_{j=1}^{m} b_{ij} u_j + \tilde{d}_i, \tag{9}$$

where:

- $e_i^{(k)}$:  $k$-th order derivative of the $i$-th component of the tracking error vector $\boldsymbol{e}(t)$ (as defined in Section 2), where $e_i(t) = g_a^{(i)}(t) - g_d^{(i)}(t)$. Here, $g_a^{(i)}(t)$ and $g_d^{(i)}(t)$ are the $i$-th components of the achieved goal $\boldsymbol{g}_a(t)$ and desired goal $\boldsymbol{g}_d(t)$, respectively.

- $u_j$: The $j$-th component of the control input deviation vector $\boldsymbol{u}(t)$, defined as $u_j = a_j - a_{d,j}$, where $a_{d,j}$ is the $j$-th component of the steady-state control action $\boldsymbol{a}_d$ required to maintain $\boldsymbol{g}_a = \boldsymbol{g}_d$.

- $a_{ijl}, b_{ij}$: Time-invariant coefficients quantifying the influence of $e_j^{(l)}$ and $u_j$ on $e_i^{(k)}$, respectively.

- $\tilde{d}_i$: Disturbance term accounting for unmodeled nonlinearities, external perturbations, and the constant offset from the steady-state action $\boldsymbol{a}_d$.

This assumption holds under the premise that the system operates within a small neighborhood of the equilibrium point (Wanner, 1995), where higher-order nonlinear terms in the Taylor expansion of the true dynamics are negligible. This local linearity enables the use of linear control theory for stability analysis and controller design.

Taking the Laplace transform of the error dynamics under zero initial conditions yields:

$$s^k E_i(s) = -\sum_{j=1}^{n} \sum_{l=0}^{k-1} a_{ijl} s^l E_j(s) + \sum_{j=1}^{m} b_{ij} U_j(s) + \tilde{D}_i(s), \tag{10}$$

where $E_i(s)$, $U_j(s)$, and $\tilde{D}_i(s)$ denote the Laplace transforms of $e_i(t)$, $u_j(t)$, and $\tilde{d}_i(t)$, respectively. This equation can be compactly expressed in matrix form as:

$$\boldsymbol{A}_{sys}(s)\boldsymbol{E}(s) = \boldsymbol{B}_{sys}\boldsymbol{U}(s) + \tilde{\boldsymbol{D}}(s), \tag{11}$$

where $\boldsymbol{A}_{sys}(s)$ is a polynomial matrix in $s$ representing the system's inherent dynamics, and $\boldsymbol{B}_{sys}$ is a constant matrix representing the input mapping.

$$\boldsymbol{A}_{sys}(s) := \begin{bmatrix} s^k + \sum_{l=0}^{k-1} a_{11l} s^l & \cdots & \sum_{l=0}^{k-1} a_{1nl} s^l \\ \vdots & \ddots & \vdots \\ \sum_{l=0}^{k-1} a_{n1l} s^l & \cdots & s^k + \sum_{l=0}^{k-1} a_{nnl} s^l \end{bmatrix} \tag{12}$$

$$\boldsymbol{B}_{sys} := \begin{bmatrix} b_{11} & \cdots & b_{1m} \\ \vdots & \ddots & \vdots \\ b_{n1} & \cdots & b_{nm} \end{bmatrix} \tag{13}$$

**Assumption B.2** (Strict Diagonal Dominance). The matrix $\boldsymbol{A}_{sys}(s)$ is strictly diagonally dominant for all $s$ in the closed right-half complex plane ($\mathrm{Re}(s) \geq 0$). That is, for every row $i = 1, 2, ..., n$, the following inequality holds:

$$|[\boldsymbol{A}_{sys}(s)]_{ii}| > \sum_{j \neq i}^{n} |[\boldsymbol{A}_{sys}(s)]_{ij}|, \quad \forall s \text{ with } \mathrm{Re}(s) \geq 0. \tag{14}$$

Assumption B.2 guarantees the invertibility of $\boldsymbol{A}_{sys}(s)$ over the region of interest, which is crucial for defining the system's transfer function. Solving for the error vector from Equation 11 yields:

$$\boldsymbol{E}(s) = \boldsymbol{A}_{sys}(s)^{-1} \left[ \boldsymbol{B}_{sys}\boldsymbol{U}(s) + \tilde{\boldsymbol{D}}(s) \right]. \tag{15}$$

This expression provides the Laplace-domain representation of the error signals in response to both the control input deviations $\boldsymbol{U}(s)$ and disturbances $\tilde{\boldsymbol{D}}(s)$.

**Assumption B.3** (Qualified Actor Policy and Linearized Inner Loop). The reinforcement learning process successfully trains an actor policy $\pi$ such that, when combined with the plant, the resulting inner control loop (from virtual goal $\boldsymbol{g}_v$ to achieved goal $\boldsymbol{g}_a$) behaves as a stable linear time-invariant system in the vicinity of the operating point. Formally, we assume that the closed-loop system formed by the linearized plant (from Assumptions A.1 and A.2) and the actor's policy (linearized around the operating point) can be represented by a transfer function matrix $\boldsymbol{G}_{\text{inner}}(s)$ from the virtual goal $\boldsymbol{G}_v(s)$ to the achieved goal $\boldsymbol{G}_a(s)$, such that $\boldsymbol{G}_a(s) = \boldsymbol{G}_{\text{inner}}(s)\boldsymbol{G}_v(s)$. This assumption implies two key properties for $\boldsymbol{G}_{\text{inner}}(s)$:

1. **Internal Stability:** All poles of $\boldsymbol{G}_{\text{inner}}(s)$ lie in the open left-half complex plane ($\mathrm{Re}(s) < 0$), ensuring that the inner loop's transient responses decay over time.

2. **Well-defined Steady-State Gain:** For a constant virtual goal $\boldsymbol{g}_{v,\text{const}}$, the achieved goal $\boldsymbol{g}_a(t)$ converges to a constant value $\boldsymbol{g}_{a,\text{const}}$ as $t \to \infty$. By the Final Value Theorem, this implies that $\lim_{s \to 0} \boldsymbol{G}_{\text{inner}}(s) = \boldsymbol{G}_{\text{inner}}(0)$ exists and is a constant matrix. Furthermore, we assume that $\boldsymbol{G}_{\text{inner}}(0)$ is invertible, meaning the actor can effectively map a non-zero virtual goal to a non-zero achieved goal in steady-state. Note that $\boldsymbol{G}_{\text{inner}}(0)$ is not necessarily an identity matrix, reflecting the potential for non-unity steady-state gain or residual errors in the actor's direct tracking of its virtual goal.

**Why RL-trained Actor Exhibits this Property?** Reinforcement learning algorithms are designed to find an optimal policy $\boldsymbol{\pi}^*$ that maximizes the expected cumulative discounted reward. This is

equivalent to solving an optimal control problem where the objective is to minimize a cost function $J(\boldsymbol{\pi})$ over an infinite horizon:

$$J(\boldsymbol{\pi}) = \mathbb{E}_{\tau \sim \boldsymbol{\pi}} \left[ \sum_{t=0}^{\infty} \gamma^t c(\boldsymbol{s}_t, \boldsymbol{a}_t) \right] \tag{16}$$

where $\tau$ is a trajectory, $\gamma \in [0, 1)$ is the discount factor, and $c(\boldsymbol{s}_t, \boldsymbol{a}_t)$ is the instantaneous cost (negative of reward). For high-precision tracking tasks, the cost function is typically engineered to heavily penalize deviations from the target. In our context, the actor's target is the virtual goal $\boldsymbol{g}_v$. A common cost structure for such tasks includes a quadratic penalty on the virtual tracking error and an action cost, consistent with the reward function mentioned in Section 2:

$$c(\boldsymbol{s}_t, \boldsymbol{a}_t) = w_e \|\boldsymbol{g}_a(\boldsymbol{s}_t) - \boldsymbol{g}_v(t)\|_2^2 + w_a \|\boldsymbol{a}_t\|_2^2 \tag{17}$$

where $w_e > 0$ and $w_a \geq 0$ are weighting factors.

To minimize $J(\boldsymbol{\pi})$, the actor's policy $\boldsymbol{\pi}$ must learn to:

1. **Minimize Virtual Tracking Error (Precision):** The dominant term in the cost function, $w_e \|\boldsymbol{g}_a(t) - \boldsymbol{g}_v(t)\|^2$, directly incentivizes the policy to drive the achieved goal $\boldsymbol{g}_a(t)$ as close as possible to the virtual goal $\boldsymbol{g}_v(t)$. A persistent non-zero virtual tracking error would lead to an unbounded cumulative cost (for $\gamma \to 1$) or a higher finite cost (for $\gamma < 1$). An optimal policy will therefore inherently converge towards driving the virtual tracking error to zero in the steady state, provided such a policy exists and the system is controllable to the virtual target. This is a fundamental property of optimal controllers designed with quadratic error costs.

2. **Maintain Stability:** Unstable behaviors, such as divergence or persistent unbounded oscillations, would lead to rapidly increasing or persistently large instantaneous costs $c(\boldsymbol{s}_t, \boldsymbol{a}_t)$. Consequently, the expected cumulative cost $J(\boldsymbol{\pi})$ for an unstable policy would be significantly higher, potentially infinite. The optimization process of RL, by minimizing this expected cost, implicitly penalizes and avoids policies that lead to instability. Therefore, successful RL training in continuous control inherently leads to policies that maintain system stability to achieve high rewards/low costs over the long run.

Therefore, this assumption posits that the outcome of a successful RL training process for a tracking task is a policy that effectively behaves as a stable and accurate inner-loop controller for its given virtual target. Even if the reward function is not precisely the quadratic form described, any reward structure designed for high-precision tracking would inherently incorporate similar principles: heavily penalizing deviations from the target and unstable behaviors. This ensures that the learned policy, by minimizing such a cost, will consistently strive for precision and stability, and its behavior can thus be approximated by an LTI system with the specified properties in the vicinity of the operating point.

**Proof of Steady-State Error Elimination**

**Theorem B.4** (Steady-State Error Elimination). *Given Assumptions B.1, B.2, and B.3, and assuming the overall closed-loop system (plant + actor + adviser) is stable, the Adviser-Actor-Critic framework with a PI adviser will eliminate steady-state errors, i.e., $\lim_{t \to \infty} \boldsymbol{e}(t) = \boldsymbol{0}$, for constant desired goals $\boldsymbol{g}_d$.*

*Proof.* We analyze the system in the Laplace domain, considering the linearized dynamics. The overall tracking error is $\boldsymbol{E}(s) = \boldsymbol{G}_a(s) - \boldsymbol{G}_d(s)$. From Assumption B.3, the inner loop (actor and plant) has a transfer function $\boldsymbol{G}_{\text{inner}}(s)$ such that $\boldsymbol{G}_a(s) = \boldsymbol{G}_{\text{inner}}(s)\boldsymbol{G}_v(s)$. The adviser's PI control law (Equation 7) in the Laplace domain is:

$$\boldsymbol{G}_v(s) = \boldsymbol{G}_d(s) - (K_p + K_i/s)\boldsymbol{E}(s) \tag{18}$$

Substitute $\boldsymbol{G}_v(s)$ into the inner loop equation:

$$\boldsymbol{G}_a(s) = \boldsymbol{G}_{\text{inner}}(s) \left[ \boldsymbol{G}_d(s) - (K_p + K_i/s)\boldsymbol{E}(s) \right] \tag{19}$$

Now substitute this expression for $\boldsymbol{G}_a(s)$ into the error definition:

$$\boldsymbol{E}(s) = \boldsymbol{G}_{\text{inner}}(s)\boldsymbol{G}_d(s) - \boldsymbol{G}_{\text{inner}}(s)(K_p + K_i/s)\boldsymbol{E}(s) - \boldsymbol{G}_d(s) \tag{20}$$

Rearranging terms to solve for $\boldsymbol{E}(s)$:

$$\boldsymbol{E}(s) + \boldsymbol{G}_{\text{inner}}(s)(K_p + K_i/s)\boldsymbol{E}(s) = \boldsymbol{G}_{\text{inner}}(s)\boldsymbol{G}_d(s) - \boldsymbol{G}_d(s) \tag{21}$$

$$[\boldsymbol{I} + \boldsymbol{G}_{\text{inner}}(s)K_p + \boldsymbol{G}_{\text{inner}}(s)K_i/s]\,\boldsymbol{E}(s) = [\boldsymbol{G}_{\text{inner}}(s) - \boldsymbol{I}]\,\boldsymbol{G}_d(s) \tag{22}$$

Thus, the closed-loop transfer function from desired goal $\boldsymbol{G}_d(s)$ to tracking error $\boldsymbol{E}(s)$ is:

$$\boldsymbol{E}(s) = [\boldsymbol{I} + \boldsymbol{G}_{\text{inner}}(s)K_p + \boldsymbol{G}_{\text{inner}}(s)K_i/s]^{-1}\,[\boldsymbol{G}_{\text{inner}}(s) - \boldsymbol{I}]\,\boldsymbol{G}_d(s) \tag{23}$$

For a constant desired goal $\boldsymbol{g}_d(t) = \boldsymbol{g}_{d,\text{const}}$, its Laplace transform is $\boldsymbol{G}_d(s) = \boldsymbol{g}_{d,\text{const}}/s$. To find the steady-state error, we apply the Final Value Theorem:

$$\lim_{t \to \infty} \boldsymbol{e}(t) = \lim_{s \to 0} s\boldsymbol{E}(s) \tag{24}$$

Substituting $\boldsymbol{G}_d(s)$ and Equation 23:

$$\lim_{t \to \infty} \boldsymbol{e}(t) = \lim_{s \to 0} s\,[\boldsymbol{I} + \boldsymbol{G}_{\text{inner}}(s)K_p + \boldsymbol{G}_{\text{inner}}(s)K_i/s]^{-1}\,[\boldsymbol{G}_{\text{inner}}(s) - \boldsymbol{I}]\,\frac{\boldsymbol{g}_{d,\text{const}}}{s} \tag{25}$$

The $s$ terms cancel:

$$\lim_{t \to \infty} \boldsymbol{e}(t) = \lim_{s \to 0} [\boldsymbol{I} + \boldsymbol{G}_{\text{inner}}(s)K_p + \boldsymbol{G}_{\text{inner}}(s)K_i/s]^{-1}\,[\boldsymbol{G}_{\text{inner}}(s) - \boldsymbol{I}]\,\boldsymbol{g}_{d,\text{const}} \tag{26}$$

Let $\boldsymbol{G}_0 = \lim_{s \to 0} \boldsymbol{G}_{\text{inner}}(s)$. From Assumption B.3, $\boldsymbol{G}_0$ exists and is invertible. Consider the term inside the inverse:

$$\lim_{s \to 0} (\boldsymbol{I} + \boldsymbol{G}_{\text{inner}}(s)K_p + \boldsymbol{G}_{\text{inner}}(s)K_i/s) = \lim_{s \to 0} (\boldsymbol{I} + \boldsymbol{G}_{\text{inner}}(s)K_p) + \lim_{s \to 0} (\boldsymbol{G}_{\text{inner}}(s)K_i/s) \tag{27}$$

The first part, $\lim_{s \to 0} (\boldsymbol{I} + \boldsymbol{G}_{\text{inner}}(s)K_p)$, converges to $\boldsymbol{I} + \boldsymbol{G}_0 K_p$, which is a finite matrix. The second part, $\lim_{s \to 0} (\boldsymbol{G}_{\text{inner}}(s)K_i/s)$, can be written as $\lim_{s \to 0} \frac{1}{s}(\boldsymbol{G}_{\text{inner}}(s)K_i)$. Since $\boldsymbol{G}_{\text{inner}}(s)$ approaches $\boldsymbol{G}_0$ as $s \to 0$, and $\boldsymbol{G}_0$ is invertible, $\boldsymbol{G}_0 K_i$ is also invertible (assuming $K_i$ is a non-singular matrix, which is typical for integral gains). Thus, this term tends to infinity (or a matrix with infinite elements) as $s \to 0$. Therefore, the entire term inside the inverse, $[\boldsymbol{I} + \boldsymbol{G}_{\text{inner}}(s)K_p + \boldsymbol{G}_{\text{inner}}(s)K_i/s]$, becomes dominated by the integral term and tends to a matrix whose inverse tends to zero:

$$\lim_{s \to 0} [\boldsymbol{I} + \boldsymbol{G}_{\text{inner}}(s)K_p + \boldsymbol{G}_{\text{inner}}(s)K_i/s]^{-1} = \boldsymbol{0} \tag{28}$$

This is because for any invertible matrix $\boldsymbol{M}$ and scalar $c$, $\lim_{s \to 0}(c/s \cdot \boldsymbol{M})^{-1} = \lim_{s \to 0}(s/c \cdot \boldsymbol{M}^{-1}) = \boldsymbol{0}$.

The second bracketed term, $[\boldsymbol{G}_{\text{inner}}(s) - \boldsymbol{I}]$, converges to a finite matrix $\boldsymbol{G}_0 - \boldsymbol{I}$ as $s \to 0$.

Combining these limits:

$$\lim_{t \to \infty} \boldsymbol{e}(t) = \boldsymbol{0} \cdot (\boldsymbol{G}_0 - \boldsymbol{I})\boldsymbol{g}_{d,\text{const}} = \boldsymbol{0} \tag{29}$$

Thus, provided the overall closed-loop system remains stable (which depends on the choice of $K_p, K_i$ and the properties of the actor and plant, as implied by Assumption B.1), the PI-based adviser ensures that the steady-state error $\boldsymbol{e}(t)$ converges to zero. $\qquad\square$

The overall closed-loop system's stability is crucial, with Assumption B.3 ensuring inner actor-plant loop stability, but the outer PI adviser loop requiring proper $K_p$ and $K_i$ tuning (Appendix E) to maintain stability and achieve desired transient response; while local linearity (Assumption B.1) aids classical control analysis, overall system stability must be verified through simulation or additional analysis beyond this steady-state error proof.

## C EXPERIMENT SETTING AND ALGORITHM CONFIGURATION

### C.1 GYMNASIUM-ROBOTICS EXPERIMENTS

To validate the proposed Adviser-Actor-Critic architecture, we conducted experiments on standard Goal-Conditioned Reinforcement Learning (GCRL) benchmarks provided by `Gymnasium-Robotics` (Plappert et al., 2018). The experimental setup utilized a computational testbed equipped with **Windows 10**, an **Intel Core i7-12700K processor**, **64GB of RAM**, and an **NVIDIA GeForce RTX 3060 Ti GPU**. Our evaluation protocol encompassed a diverse range of robotic control challenges, specifically including Fetch manipulation tasks and ShadowHand dexterous manipulation (as illustrated in Figure 3), thereby ensuring comprehensive coverage of various multi-goal reinforcement learning paradigms.

Table 4: Detailed Hyperparameters for Soft Actor-Critic (SAC)

| Parameter | Value |
|---|---|
| **Optimization Settings** | |
| Actor Learning Rate | $3 \times 10^{-4}$ |
| Critic Learning Rate | $3 \times 10^{-4}$ |
| Temperature ($\alpha$) Learning Rate | $3 \times 10^{-4}$ |
| Optimizer | Adam |
| Discount Factor ($\gamma$) | 0.99 |
| Target Network Update Polyak Coefficient ($\tau$) | 0.005 |
| Gradient Steps per Environment Step | 1 |
| **Replay Buffer Configuration** | |
| Replay Buffer Capacity | $10^6$ transitions |
| Batch Size | 256 |
| **Policy Network (Actor) Architecture** | |
| Actor Network Layers | 2 |
| Actor Units per Layer | 256 |
| Actor Hidden Layer Activation | ReLU |
| Actor Output Layer Activation | Tanh (for continuous action spaces) |
| Log Standard Deviation (Min/Max) | $[-20, 2]$ |
| **Q-Function Network (Critic) Architecture** | |
| Critic Network Layers | 2 |
| Critic Units per Layer | 256 |
| Critic Hidden Layer Activation | ReLU |
| **Entropy Regularization** | |
| Target Entropy | Automatic (set to $-\dim(\text{action space})$) |

Table 5: Detailed Hyperparameters for Proximal Policy Optimization (PPO)

| Parameter | Value |
|---|---|
| **Optimization Settings** | |
| Learning Rate | $3 \times 10^{-4}$ |
| Optimizer | Adam |
| Adam Epsilon ($\epsilon_{\text{Adam}}$) | $1 \times 10^{-5}$ |
| Discount Factor ($\gamma$) | 0.99 |
| Generalized Advantage Estimation (GAE) Lambda ($\lambda$) | 0.95 |
| **Policy Update Configuration** | |
| PPO Clipping Epsilon ($\epsilon_{\text{clip}}$) | 0.2 |
| Entropy Regularization Coefficient | 0.01 |
| Value Function Loss Coefficient | 0.5 |
| Number of Epochs per Policy Update | 10 |
| Minibatch Size | 64 |
| Rollout Length (Environment Steps per Update) | 2048 |
| **Network Architecture** | |
| Actor Network Layers | 2 |
| Actor Units per Layer | 64 |
| Actor Hidden Layer Activation | Tanh |
| Critic Network Layers | 2 |
| Critic Units per Layer | 64 |
| Critic Hidden Layer Activation | Tanh |

Table 6: Detailed Hyperparameters for DreamerV3

| Parameter | Value |
|---|---|
| **Optimization Settings** | |
| World Model Learning Rate | $1 \times 10^{-4}$ |
| Actor Learning Rate | $3 \times 10^{-5}$ |
| Critic Learning Rate | $3 \times 10^{-5}$ |
| Optimizer | Adam |
| Adam Epsilon ($\epsilon_{\text{Adam}}$) | $1 \times 10^{-8}$ |
| Discount Factor ($\gamma$) | 0.99 |
| Lambda ($\lambda$) | 0.95 |
| **World Model Configuration** | |
| Recurrent State-Space Model (RSSM) Stochastic State Size | 32 |
| RSSM Deterministic State Size | 256 |
| RSSM Hidden Layer Size | 256 |
| RSSM Number of Layers | 1 |
| Encoder/Decoder Number of Layers | 4 |
| Encoder/Decoder Units per Layer | 256 |
| Reward Head Number of Layers | 2 |
| Reward Head Units per Layer | 256 |
| Continue Head Number of Layers | 2 |
| Continue Head Units per Layer | 256 |
| Free Bits (KL Divergence Loss) | 1.0 |
| KL Divergence Loss Weight | 1.0 |
| Reward Loss Weight | 1.0 |
| Continue Loss Weight | 1.0 |
| **Agent (Actor-Critic) Configuration** | |
| Actor Network Layers | 4 |
| Actor Units per Layer | 256 |
| Critic Network Layers | 4 |
| Critic Units per Layer | 256 |
| Imagination Horizon (Steps) | 15 |
| **Training Parameters** | |
| Batch Size | 16 |
| Sequence Length | 50 |
| Gradient Clipping Norm | 100.0 |

Table 7: Configuration for Wide Normalization (WN)

| Parameter | Value |
|---|---|
| **Network Architecture** | |
| Actor Network Layers | 2 |
| Actor Units per Layer | 256 |
| Actor Hidden Layer Activation | ReLU |
| Actor Output Layer Activation | Tanh (for continuous actions) |
| Critic Network Layers | 2 |
| Critic Units per Layer | 256 |
| Critic Hidden Layer Activation | ReLU |
| Encoder Type $(e_1, e_2)$ | Dual Multi-Layer Perceptrons (MLPs) |
| Encoder Number of Layers | 2 |
| Encoder Units per Layer | 176 |
| Encoder Hidden Layer Activation | ReLU |
| Specific Implementation | $\ell_2$-constrained residual layers |
| **Optimization Settings** | |
| Actor Learning Rate | $1 \times 10^{-4}$ |
| Critic Learning Rate | $1 \times 10^{-4}$ |
| Optimizer | Adam |
| Discount Factor $(\gamma)$ | 0.99 |
| Target Network Update Polyak Coefficient $(\tau)$ | 0.005 |
| Gradient Steps per Environment Step | 1 |
| **Replay Buffer Configuration** | |
| Replay Buffer Capacity | $10^6$ transitions |
| Batch Size | 256 |
| HER Strategy | `future` |
| HER Ratio | 0.8 |
| HER Goal Sampling Number | 4 |

Table 8: Configuration for Metric Residual Network (MRN)

| Parameter | Value |
|---|---|
| **Network Architecture** | |
| Actor Network Layers | 2 |
| Actor Units per Layer | 256 |
| Actor Hidden Layer Activation | ReLU |
| Actor Output Layer Activation | Tanh (for continuous actions) |
| Critic Network Layers | 2 |
| Critic Units per Layer | 256 |
| Critic Hidden Layer Activation | ReLU |
| Encoder Type ($e_1, e_2$) | Dual Multi-Layer Perceptrons (MLPs) |
| Encoder Number of Layers | 2 |
| Encoder Units per Layer | 176 |
| Encoder Hidden Layer Activation | ReLU |
| Specific Implementation | $\ell_2$-constrained residual layers |
| **Optimization Settings** | |
| Actor Learning Rate | $1 \times 10^{-4}$ |
| Critic Learning Rate | $1 \times 10^{-4}$ |
| Optimizer | Adam |
| Discount Factor ($\gamma$) | 0.99 |
| Target Network Update Polyak Coefficient ($\tau$) | 0.005 |
| Gradient Steps per Environment Step | 1 |
| **Replay Buffer Configuration** | |
| Replay Buffer Capacity | $10^6$ transitions |
| Batch Size | 256 |
| HER Strategy | `future` |
| HER Ratio | 0.8 |
| HER Goal Sampling Number | 4 |

## C.2 REAL-WORLD QUADCOPTER EXPERIMENTS

The quadcopter control system implements a simulation-to-reality pipeline based on (Bousbaine et al., 2012). For convenience, we employ a reinforcement learning controller trained in our laboratory in earlier work, which uses different parameters than the SAC implementation detailed earlier. Virtual training employs a 3-DOF orientation dynamics model within Gazebo/ROS environment, interfaced through Python-based reinforcement learning framework. Physical deployment executes optimized policies on Raspberry Pi 4B (8GB RAM) coordinating with Pixhawk4 flight controller, achieving 200Hz PWM motor control via hardware-timed GPIO. The airframe features 35.0 cm axle distance with total mass 1.40 kg.

Table 9: Hyperparameters for Quadcopter Control Experiments (SAC Framework)

| Parameter | Specification |
|---|---|
| **Algorithm Implementation** | |
| Base framework | Soft Actor-Critic (SAC) |
| **Network Architecture** | |
| Hidden layers | 3 fully-connected layers (128 units/layer) |
| Activation function | Self-normalizing SELU units |
| **Optimization Protocol** | |
| Q-network learning rate | $5 \times 10^{-4}$ (Adam optimizer) |
| Policy learning rate | $3 \times 10^{-4}$ (Adam optimizer) |
| Temperature ($\alpha$) | Initial value 0.2 with learning rate $3 \times 10^{-4}$ |
| **Value Estimation** | |
| Discount factor | $\gamma = 0.995$ |
| Soft update rate | $\tau = 0.005$ |
| **Experience Management** | |
| Replay buffer | Capacity $1 \times 10^6$ transitions |
| Warmup period | 1,000 initial samples |
| Batch configuration | Minibatch size 64 samples/update |
| **Training Protocol** | |
| Training duration | 1,000 optimization epochs |

# D    COMPARISON EXPERIMENTS RESULTS

This section evaluates the Adviser-Actor-Critic (AAC) framework's performance in steady-state error reduction and success rates across six robotic control tasks. We compare AAC with five baseline methods—Metric Residual Network (MRN) (Liu et al., 2023), Wide Normalization (WN) (Pitis et al., 2019), SAC(He et al., 2020), PPO(Schulman et al., 2017) and DreamerV3(Hafner et al., 2025)—using static error metrics (bar charts) and trajectory analysis (line charts) on six Gymnasium-Robotics benchmark environments. These visualizations demonstrate AAC's superiority in systematic steady-state error mitigation and improved success rates (44-hour runtime).

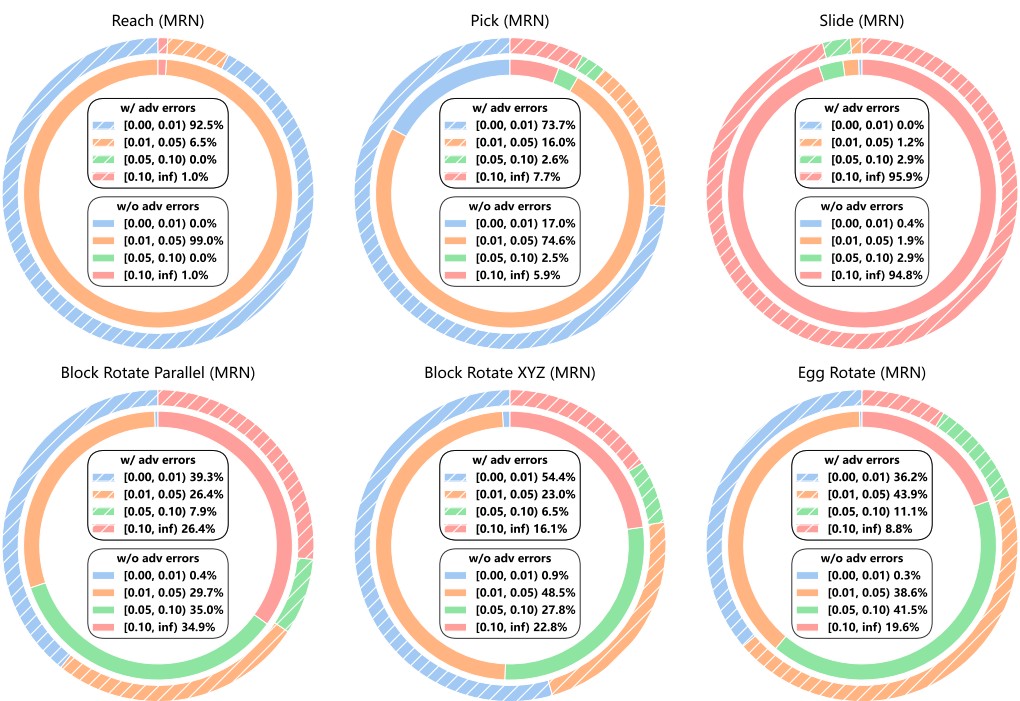

Figure 8: Steady-State Error Comparison: AAC vs. MRN in 6 Benchmark Environments (Bar Chart)

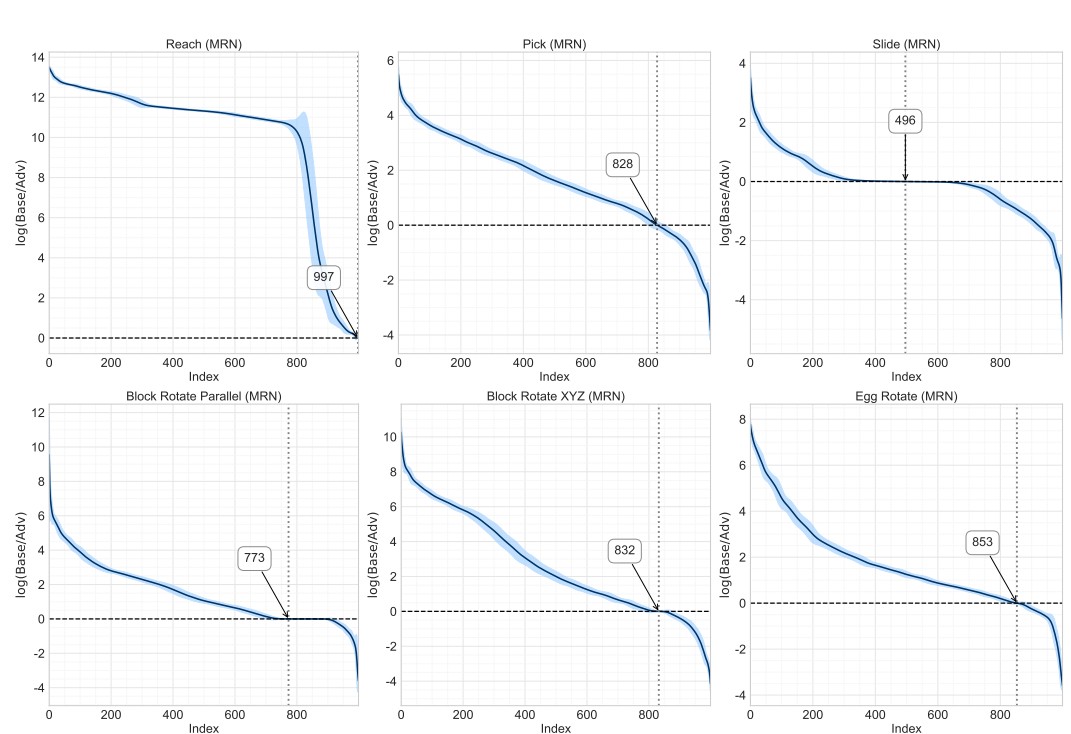

Figure 9: Steady-State Error Trajectory: AAC vs. MRN in 6 Benchmark Environments (Line Chart)

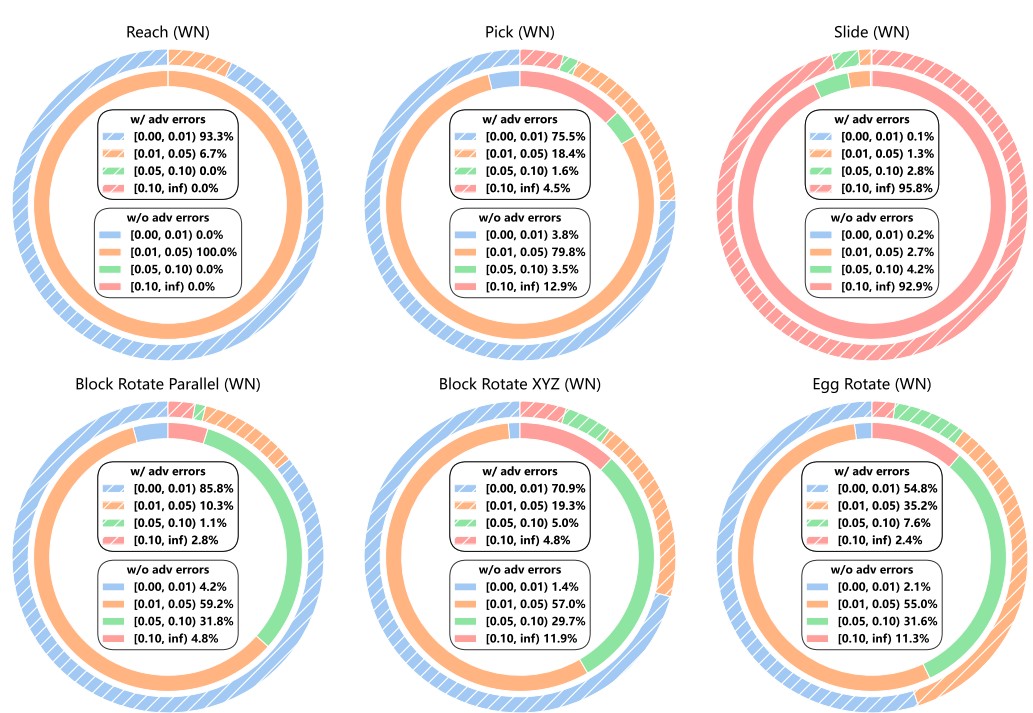

Figure 10: Steady-State Error Comparison: AAC vs. WN in 6 Benchmark Environments (Bar Chart)

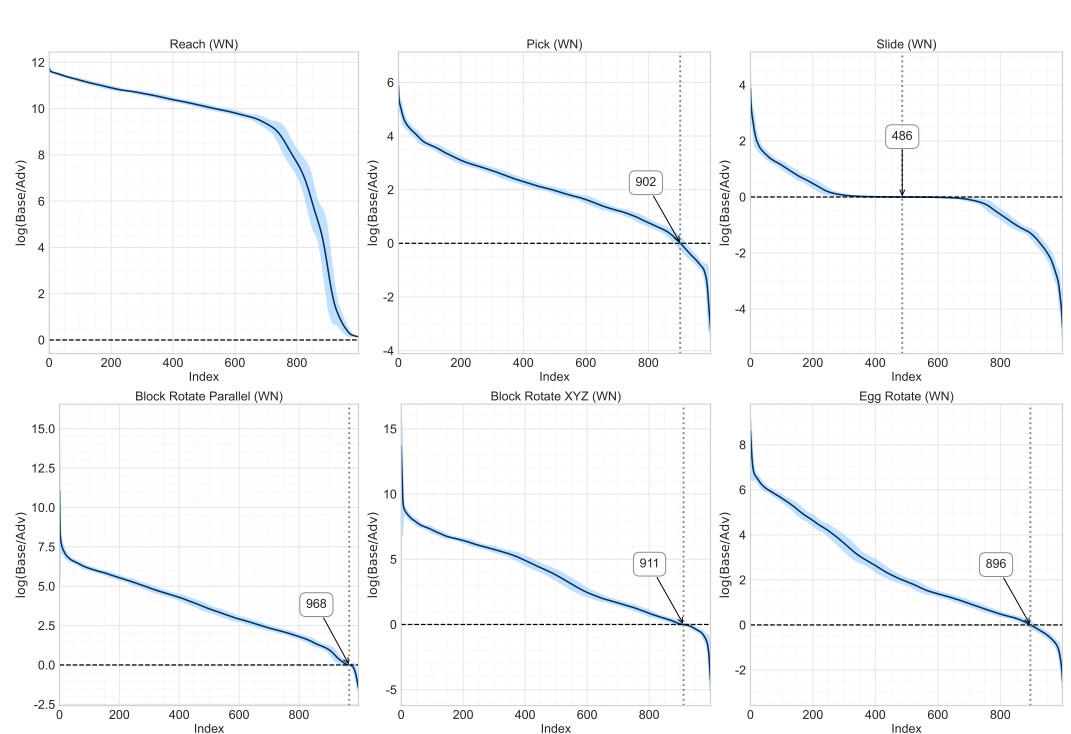

Figure 11: Steady-State Error Trajectory: AAC vs. WN in 6 Benchmark Environments (Line Chart)

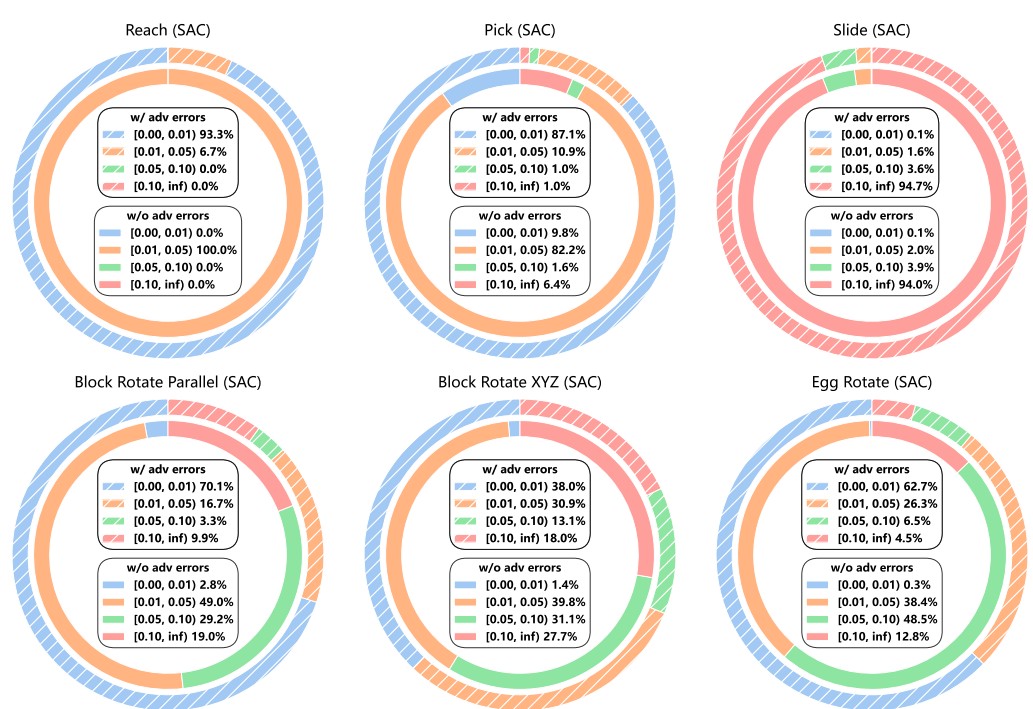

Figure 12: Steady-State Error Comparison: AAC vs. SAC in 6 Benchmark Environments (Bar Chart)

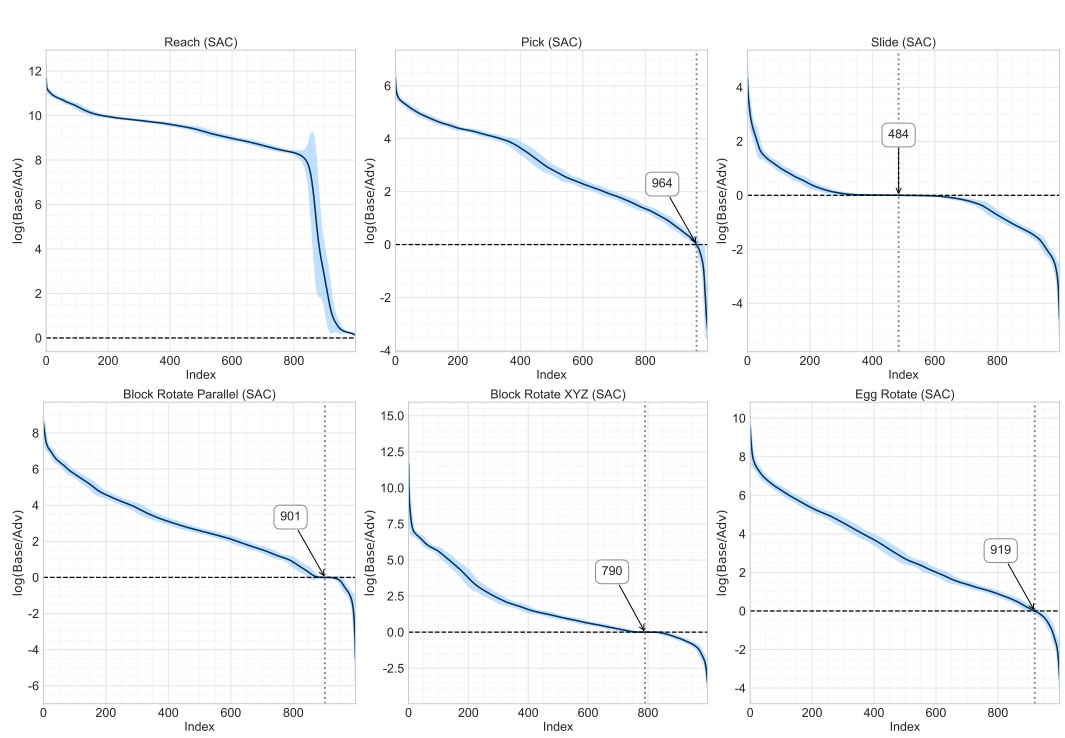

Figure 13: Steady-State Error Trajectory: AAC vs. SAC in 6 Benchmark Environments (Line Chart)

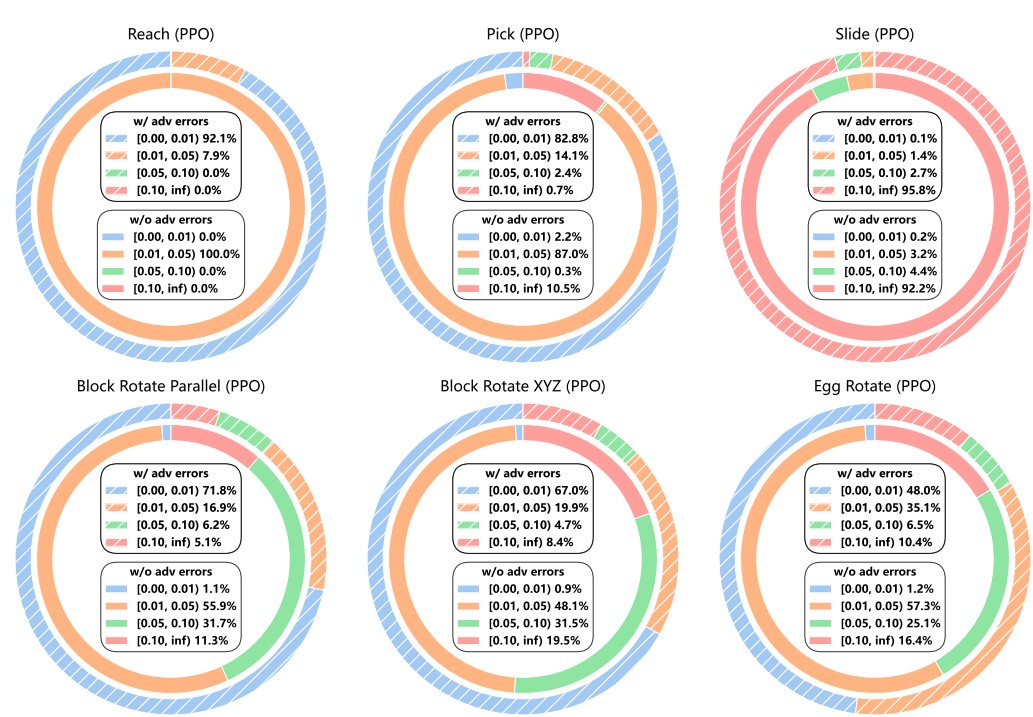

Figure 14: Steady-State Error Comparison: AAC vs. PPO in 6 Benchmark Environments (Bar Chart)

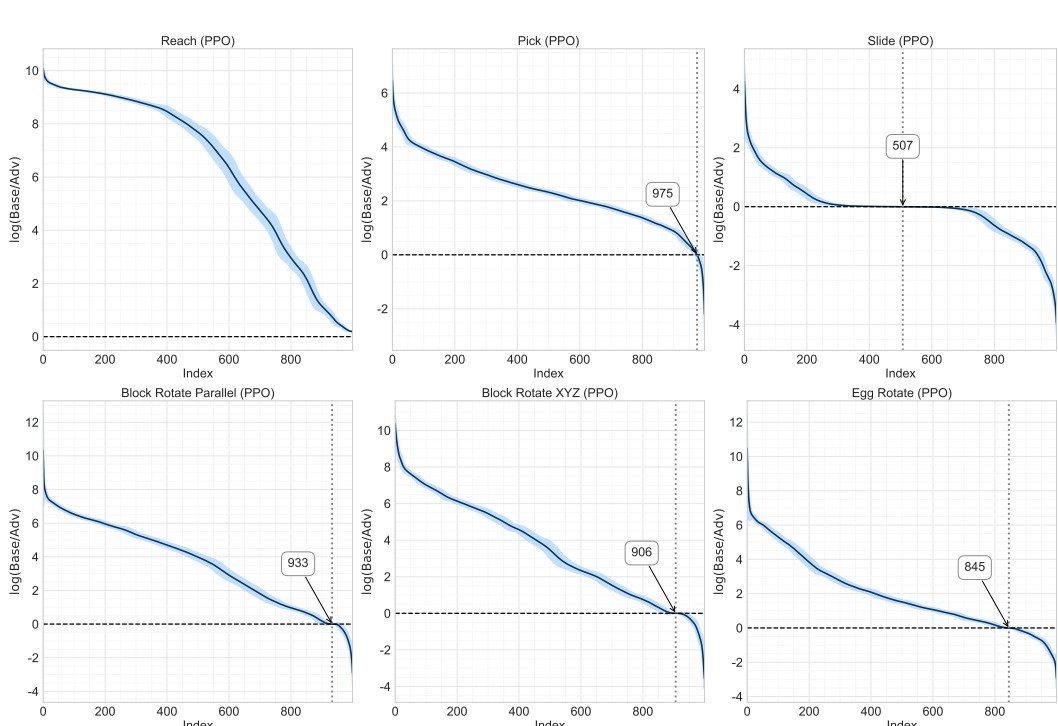

Figure 15: Steady-State Error Trajectory: AAC vs. PPO in 6 Benchmark Environments (Line Chart)

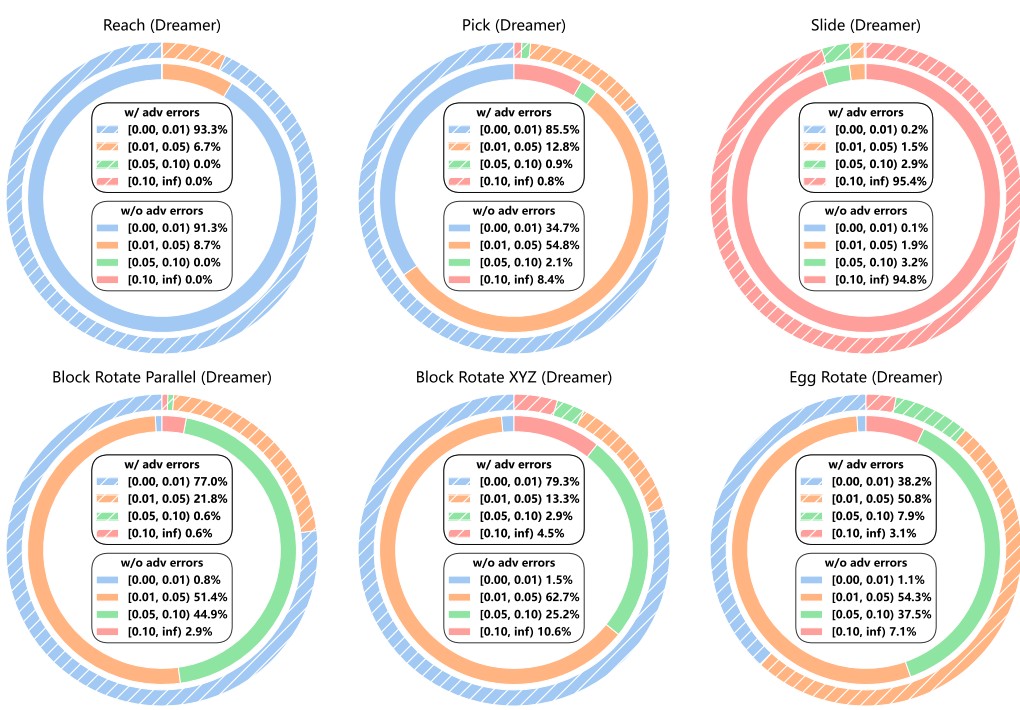

Figure 16: Steady-State Error Comparison: AAC vs. DreamerV3 in 6 Benchmark Environments (Bar Chart)

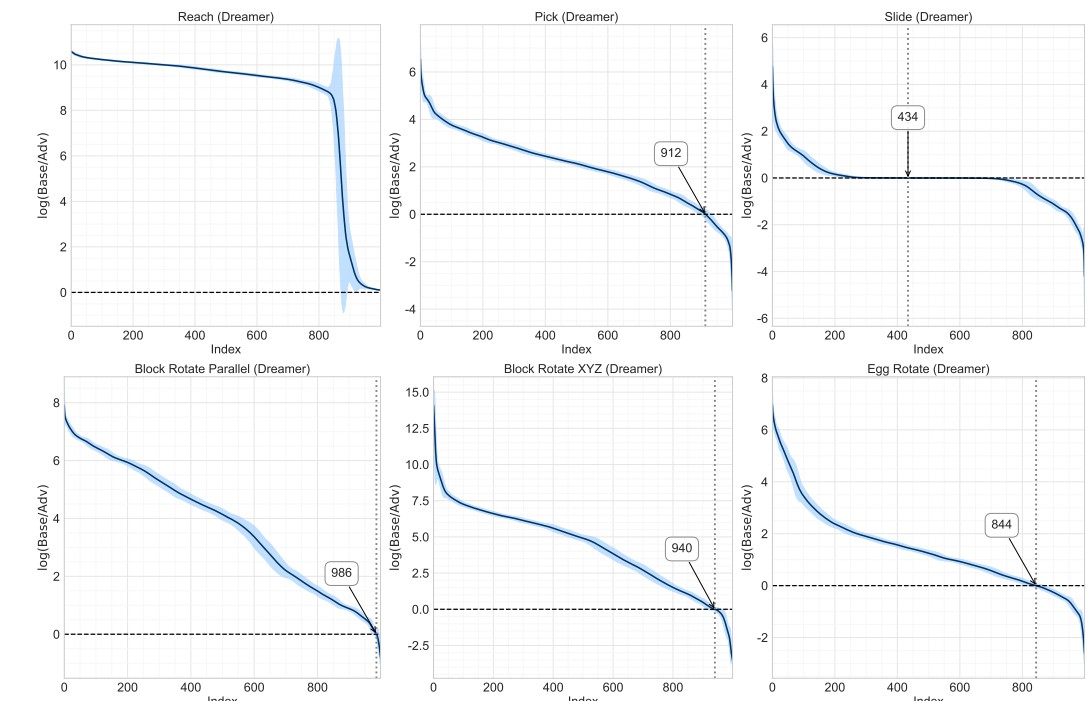

Figure 17: Steady-State Error Trajectory: AAC vs. DreamerV3 in 6 Benchmark Environments (Line Chart)

## E  ROBUSTNESS ANALYSIS OF PARAMETRIC EFFECTS

This section presents a systematic investigation into the influence of critical control parameters—$K_i$ (P1) and $\sigma$ (P2)—on three performance indices: Steady-State Error, ITAE, and Success Rate. Through factorial design experiments coupled with ANOVA analysis, statistically significant parametric effects ($p < 0.01$) are identified for a majority of performance metrics across various methods and tasks, establishing their criticality in system behavior modulation. However, some exceptions are observed, particularly for certain metrics in tasks like Block Rotate Parallel and Block Rotate XYZ, and for PPO's success rate in the Reach task, where parameter effects were not statistically significant at the $p < 0.01$ level. Notably, the system exhibits exceptional tolerance to parameter perturbations, with $\pm 30\%$ deviations from optimal values producing less than 5% degradation in performance measures. This dual finding—parametric significance combined with robust performance under variation—demonstrates the adaptive controller's favorable trade-off between sensitivity to intentional tuning and insensitivity to unintended parameter drifts. The results substantiate the practical feasibility of automated tuning protocols in robotic implementations, where environmental variability and component aging necessitate robust parameter operating ranges. It takes 15 hours to run.

**Definition of Integral Time Absolute Error (ITAE)**:

$$\text{ITAE} = \int_0^T t \cdot \|\boldsymbol{e}\|_1 \, dt \tag{30}$$

where $\boldsymbol{e} := \boldsymbol{g}_d - \boldsymbol{g}_a \in \mathbb{R}^n$ represents the error vector at time $t$, $\|\cdot\|_1$ denotes the $L_1$-norm, and $T$ corresponds to the evaluation time horizon. The ITAE metric effectively captures both temporal error persistence and magnitude accumulation through its weighted integral formulation, which emphasizes prolonged error conditions more strongly than instantaneous errors.

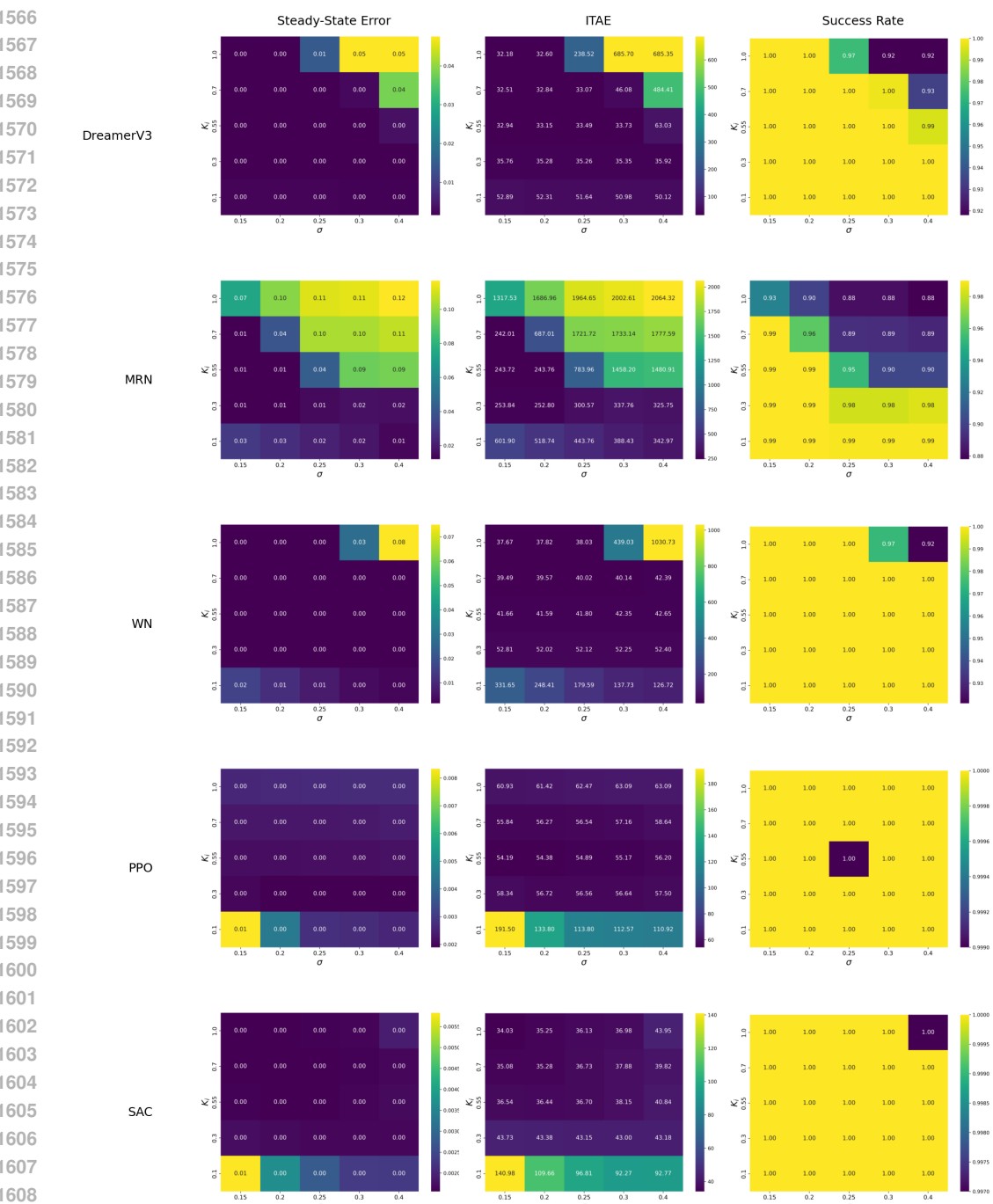

Figure 18: Heatmap Visualization of Method Performance Across Metrics and Hyperparameters for FetchReach Task

Table 10: Two-way ANOVA Results for FetchReach

| Baseline | Metric | P1 (F) | P1 (p) | P2 (F) | P2 (p) | Int. (F) | Int. (p) |
|---|---|---|---|---|---|---|---|
| SAC | Steady-State Error | 95.57 | **0.000** | 12.71 | **0.000** | 14.37 | **0.000** |
| | ITAE | 362.43 | **0.000** | 4.53 | **0.001** | 7.57 | **0.000** |
| | Success Rate | 3.01 | 0.017 | 3.01 | 0.017 | 3.01 | **0.000** |
| PPO | Steady-State Error | 123.67 | **0.000** | 36.02 | **0.000** | 37.12 | **0.000** |
| | ITAE | 460.09 | **0.000** | 17.63 | **0.000** | 19.69 | **0.000** |
| | Success Rate | 1.00 | 0.406 | 1.00 | 0.406 | 1.00 | 0.453 |
| DreamerV3 | Steady-State Error | 118.66 | **0.000** | 77.44 | **0.000** | 39.98 | **0.000** |
| | ITAE | 108.28 | **0.000** | 64.79 | **0.000** | 34.14 | **0.000** |
| | Success Rate | 137.47 | **0.000** | 88.40 | **0.000** | 44.04 | **0.000** |
| WN | Steady-State Error | 276.89 | **0.000** | 60.25 | **0.000** | 59.26 | **0.000** |
| | ITAE | 578.35 | **0.000** | 36.37 | **0.000** | 36.05 | **0.000** |
| | Success Rate | 0.00 | 1.000 | 0.00 | 1.000 | 0.00 | 1.000 |
| MRN | Steady-State Error | 51.66 | **0.000** | 19.46 | **0.000** | 27.39 | **0.000** |
| | ITAE | 46.88 | **0.000** | 19.46 | **0.000** | 23.03 | **0.000** |
| | Success Rate | 41.60 | **0.000** | 19.83 | **0.000** | 19.83 | **0.000** |

Table 11: Two-way ANOVA Results for FetchPick

| Baseline | Metric | P1 (F) | P1 (p) | P2 (F) | P2 (p) | Int. (F) | Int. (p) |
|---|---|---|---|---|---|---|---|
| SAC | Steady-State Error | 24.05 | **0.000** | 3.47 | **0.008** | 2.26 | **0.003** |
| | ITAE | 71.43 | **0.000** | 8.16 | **0.000** | 4.48 | **0.000** |
| | Success Rate | 15.42 | **0.000** | 2.92 | 0.020 | 2.16 | **0.005** |
| PPO | Steady-State Error | 99.66 | **0.000** | 5.94 | **0.000** | 5.41 | **0.000** |
| | ITAE | 167.93 | **0.000** | 6.75 | **0.000** | 7.21 | **0.000** |
| | Success Rate | 32.28 | **0.000** | 1.99 | 0.093 | 1.71 | 0.038 |
| DreamerV3 | Steady-State Error | 42.26 | **0.000** | 1.58 | 0.177 | 4.64 | **0.000** |
| | ITAE | 93.47 | **0.000** | 4.37 | **0.002** | 6.90 | **0.000** |
| | Success Rate | 46.39 | **0.000** | 2.05 | 0.085 | 3.29 | **0.000** |
| WN | Steady-State Error | 7.99 | **0.000** | 0.11 | 0.980 | 1.89 | 0.017 |
| | ITAE | 31.98 | **0.000** | 0.79 | 0.532 | 2.29 | **0.002** |
| | Success Rate | 8.80 | **0.000** | 0.24 | 0.916 | 1.40 | 0.130 |
| MRN | Steady-State Error | 18.03 | **0.000** | 1.85 | 0.116 | 2.89 | **0.000** |
| | ITAE | 18.47 | **0.000** | 2.97 | 0.018 | 2.66 | **0.000** |
| | Success Rate | 12.13 | **0.000** | 1.10 | 0.353 | 0.58 | 0.903 |

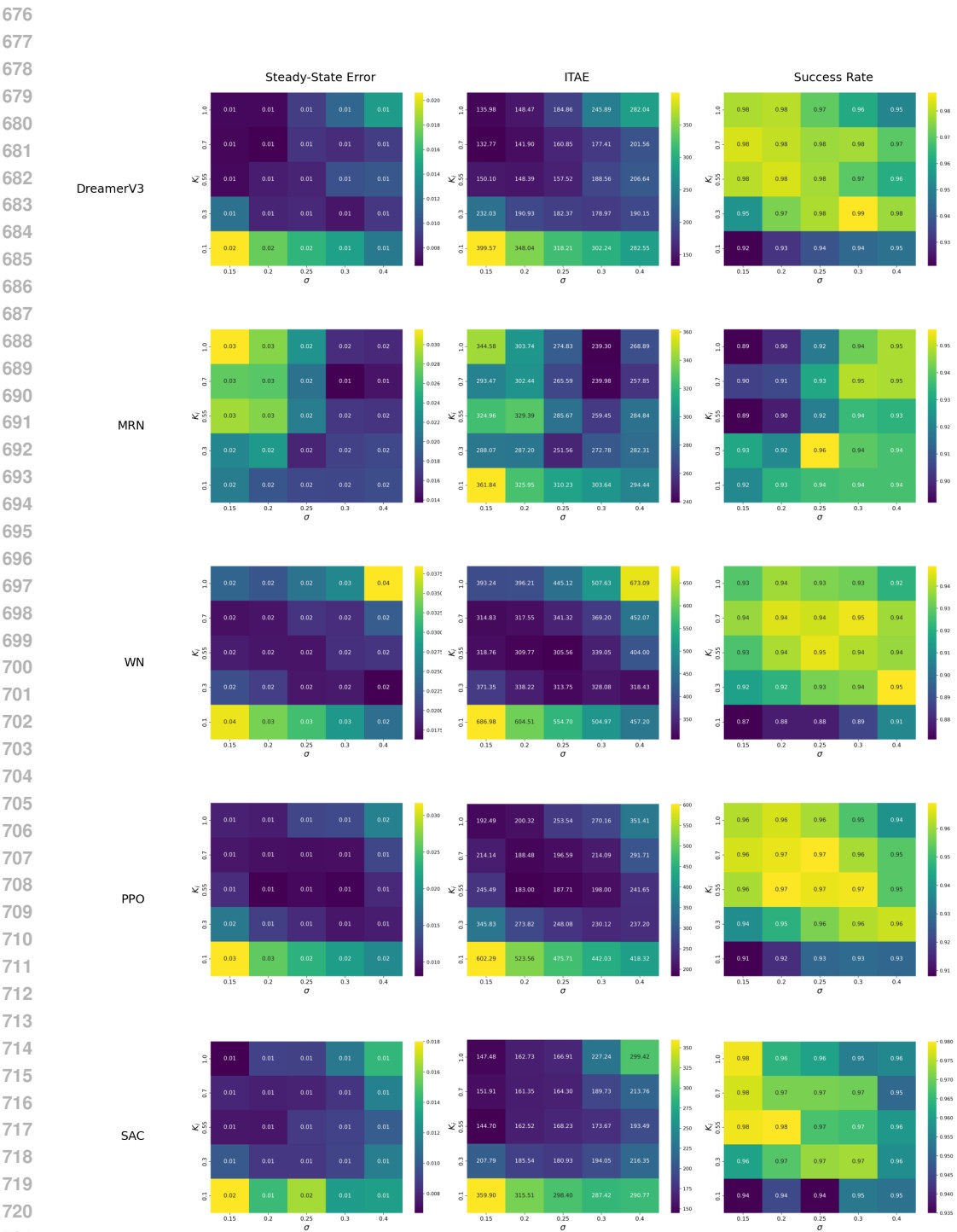

Figure 19: Heatmap Visualization of Method Performance Across Metrics and Hyperparameters for FetchPick Task

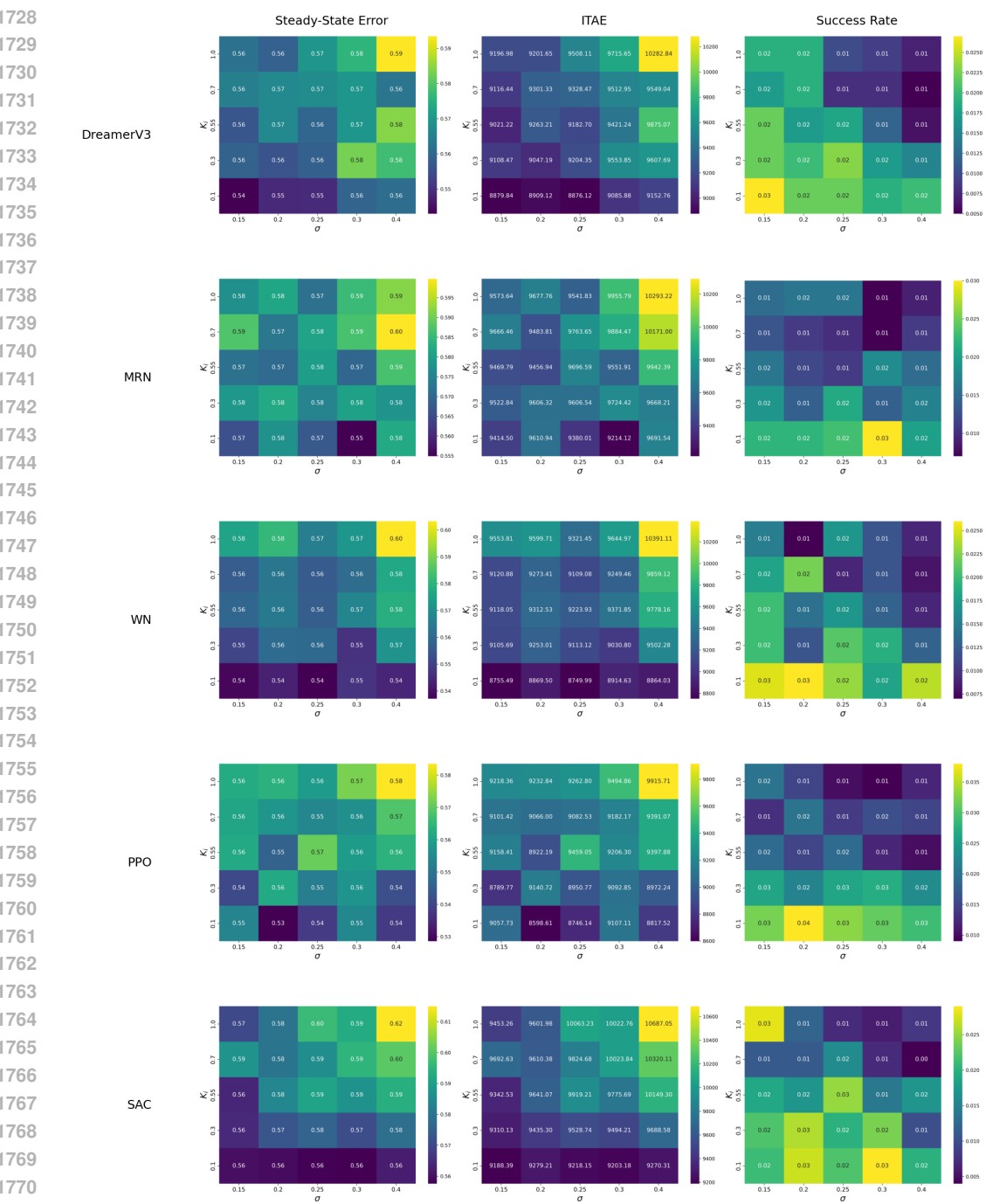

Figure 20: Heatmap Visualization of Method Performance Across Metrics and Hyperparameters for FetchSlide Task

Table 12: Two-way ANOVA Results for FetchSlide

| Baseline | Metric | P1 (F) | P1 (p) | P2 (F) | P2 (p) | Int. (F) | Int. (p) |
|---|---|---|---|---|---|---|---|
| SAC | Steady-State Error | 13.15 | **0.000** | 4.54 | **0.001** | 0.86 | 0.619 |
| | ITAE | 19.98 | **0.000** | 12.19 | **0.000** | 1.53 | 0.078 |
| | Success Rate | 10.14 | **0.000** | 3.15 | 0.013 | 1.85 | 0.020 |
| PPO | Steady-State Error | 7.79 | **0.000** | 0.93 | 0.444 | 1.48 | 0.096 |
| | ITAE | 10.80 | **0.000** | 3.45 | **0.008** | 1.62 | 0.055 |
| | Success Rate | 17.19 | **0.000** | 0.87 | 0.483 | 0.54 | 0.930 |
| DreamerV3 | Steady-State Error | 4.32 | **0.002** | 5.29 | **0.000** | 0.75 | 0.746 |
| | ITAE | 10.67 | **0.000** | 15.16 | **0.000** | 1.05 | 0.394 |
| | Success Rate | 6.30 | **0.000** | 6.22 | **0.000** | 0.50 | 0.948 |
| WN | Steady-State Error | 5.31 | **0.000** | 1.19 | 0.315 | 0.69 | 0.803 |
| | ITAE | 5.03 | **0.000** | 1.70 | 0.147 | 0.55 | 0.923 |
| | Success Rate | 13.64 | **0.000** | 0.89 | 0.469 | 0.67 | 0.830 |
| MRN | Steady-State Error | 6.52 | **0.000** | 3.62 | **0.006** | 0.92 | 0.542 |
| | ITAE | 9.76 | **0.000** | 6.83 | **0.000** | 1.43 | 0.115 |
| | Success Rate | 4.16 | **0.002** | 0.95 | 0.435 | 0.38 | 0.987 |

Table 13: Two-way ANOVA Results for HandManipulateBlockRotateParallel

| Baseline | Metric | P1 (F) | P1 (p) | P2 (F) | P2 (p) | Int. (F) | Int. (p) |
|---|---|---|---|---|---|---|---|
| SAC | Steady-State Error | 2.44 | 0.045 | 0.49 | 0.745 | 0.05 | 1.000 |
| | ITAE | 2.30 | 0.056 | 0.49 | 0.745 | 0.04 | 1.000 |
| | Success Rate | 16.04 | **0.000** | 6.58 | **0.000** | 0.20 | 1.000 |
| PPO | Steady-State Error | 2.23 | 0.064 | 0.09 | 0.984 | 0.01 | 1.000 |
| | ITAE | 2.12 | 0.075 | 0.06 | 0.992 | 0.01 | 1.000 |
| | Success Rate | 11.64 | **0.000** | 0.73 | 0.575 | 0.33 | 0.994 |
| DreamerV3 | Steady-State Error | 85.94 | **0.000** | 5.09 | **0.000** | 0.63 | 0.864 |
| | ITAE | 57.76 | **0.000** | 2.53 | 0.039 | 0.28 | 0.998 |
| | Success Rate | 5.06 | **0.000** | 0.41 | 0.803 | 0.72 | 0.771 |
| WN | Steady-State Error | 4.52 | **0.001** | 0.27 | 0.898 | 0.16 | 1.000 |
| | ITAE | 3.65 | **0.006** | 0.14 | 0.966 | 0.13 | 1.000 |
| | Success Rate | 4.50 | **0.001** | 0.17 | 0.954 | 0.48 | 0.956 |
| MRN | Steady-State Error | 2.38 | 0.049 | 0.11 | 0.978 | 0.02 | 1.000 |
| | ITAE | 2.30 | 0.056 | 0.08 | 0.989 | 0.01 | 1.000 |
| | Success Rate | 2.53 | 0.038 | 0.54 | 0.705 | 0.42 | 0.978 |

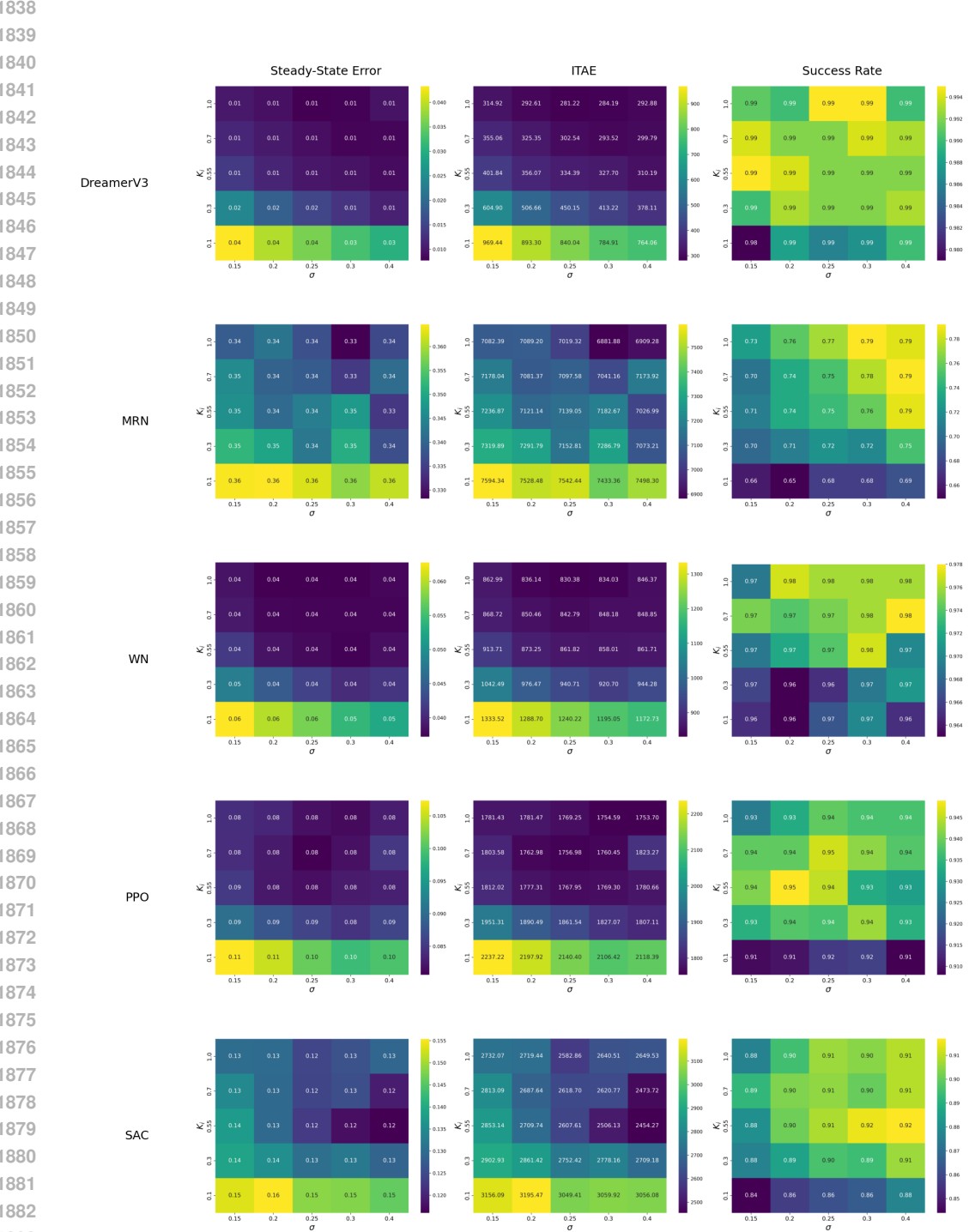

Figure 21: Combined Heatmap Visualization of Method Performance Across Metrics and Hyperparameters for BlockRotateParallel Task

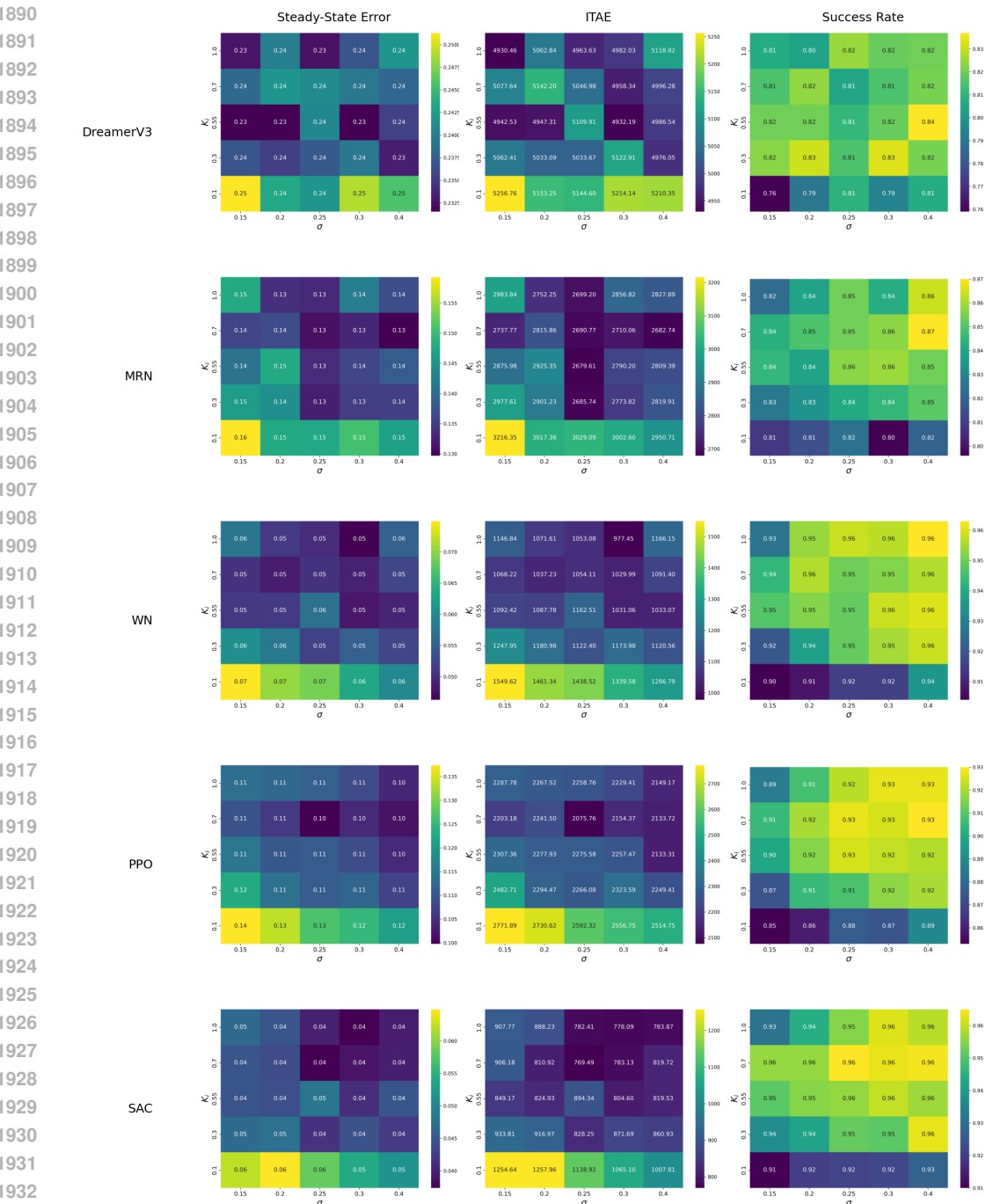

Figure 22: Combined Heatmap Visualization of Method Performance Across Metrics and Hyperparameters for HandManipulateBlockRotateXYZ Task

Table 14: Two-way ANOVA Results for HandManipulateBlockRotateXYZ

| Baseline | Metric | P1 (F) | P1 (p) | P2 (F) | P2 (p) | Int. (F) | Int. (p) |
|---|---|---|---|---|---|---|---|
| SAC | Steady-State Error | 4.60 | **0.001** | 0.77 | 0.546 | 0.13 | 1.000 |
| | ITAE | 5.44 | **0.000** | 0.72 | 0.576 | 0.14 | 1.000 |
| | Success Rate | 25.35 | **0.000** | 4.79 | **0.001** | 0.46 | 0.964 |
| PPO | Steady-State Error | 2.19 | 0.067 | 0.51 | 0.728 | 0.03 | 1.000 |
| | ITAE | 2.19 | 0.068 | 0.29 | 0.883 | 0.03 | 1.000 |
| | Success Rate | 25.21 | **0.000** | 11.06 | **0.000** | 0.40 | 0.982 |
| DreamerV3 | Steady-State Error | 0.22 | 0.928 | 0.00 | 1.000 | 0.03 | 1.000 |
| | ITAE | 0.17 | 0.955 | 0.00 | 1.000 | 0.03 | 1.000 |
| | Success Rate | 5.49 | **0.000** | 1.17 | 0.324 | 0.83 | 0.647 |
| WN | Steady-State Error | 1.29 | 0.272 | 0.27 | 0.900 | 0.05 | 1.000 |
| | ITAE | 1.52 | 0.192 | 0.12 | 0.973 | 0.03 | 1.000 |
| | Success Rate | 12.22 | **0.000** | 2.92 | 0.020 | 0.48 | 0.957 |
| MRN | Steady-State Error | 1.47 | 0.207 | 0.20 | 0.938 | 0.14 | 1.000 |
| | ITAE | 1.53 | 0.190 | 0.19 | 0.942 | 0.11 | 1.000 |
| | Success Rate | 13.63 | **0.000** | 4.92 | **0.001** | 0.34 | 0.993 |

Table 15: Two-way ANOVA Results for HandManipulateEggRotate

| Baseline | Metric | P1 (F) | P1 (p) | P2 (F) | P2 (p) | Int. (F) | Int. (p) |
|---|---|---|---|---|---|---|---|
| SAC | Steady-State Error | 15.72 | **0.000** | 0.33 | 0.856 | 0.45 | 0.969 |
| | ITAE | 23.08 | **0.000** | 0.25 | 0.908 | 0.32 | 0.995 |
| | Success Rate | 10.88 | **0.000** | 0.86 | 0.487 | 0.91 | 0.559 |
| PPO | Steady-State Error | 2.56 | 0.037 | 0.14 | 0.966 | 0.05 | 1.000 |
| | ITAE | 2.62 | 0.033 | 0.11 | 0.981 | 0.03 | 1.000 |
| | Success Rate | 10.78 | **0.000** | 2.60 | 0.034 | 0.56 | 0.913 |
| DreamerV3 | Steady-State Error | 4.02 | **0.003** | 0.17 | 0.952 | 0.06 | 1.000 |
| | ITAE | 3.91 | **0.004** | 0.16 | 0.958 | 0.05 | 1.000 |
| | Success Rate | 4.85 | **0.001** | 0.65 | 0.626 | 0.58 | 0.903 |
| WN | Steady-State Error | 0.12 | 0.976 | 0.01 | 1.000 | 0.01 | 1.000 |
| | ITAE | 0.12 | 0.974 | 0.00 | 1.000 | 0.00 | 1.000 |
| | Success Rate | 1.00 | 0.407 | 0.38 | 0.823 | 0.23 | 0.999 |
| MRN | Steady-State Error | 4.21 | **0.002** | 0.24 | 0.916 | 0.06 | 1.000 |
| | ITAE | 4.19 | **0.002** | 0.20 | 0.937 | 0.04 | 1.000 |
| | Success Rate | 32.28 | **0.000** | 2.87 | 0.022 | 0.45 | 0.970 |

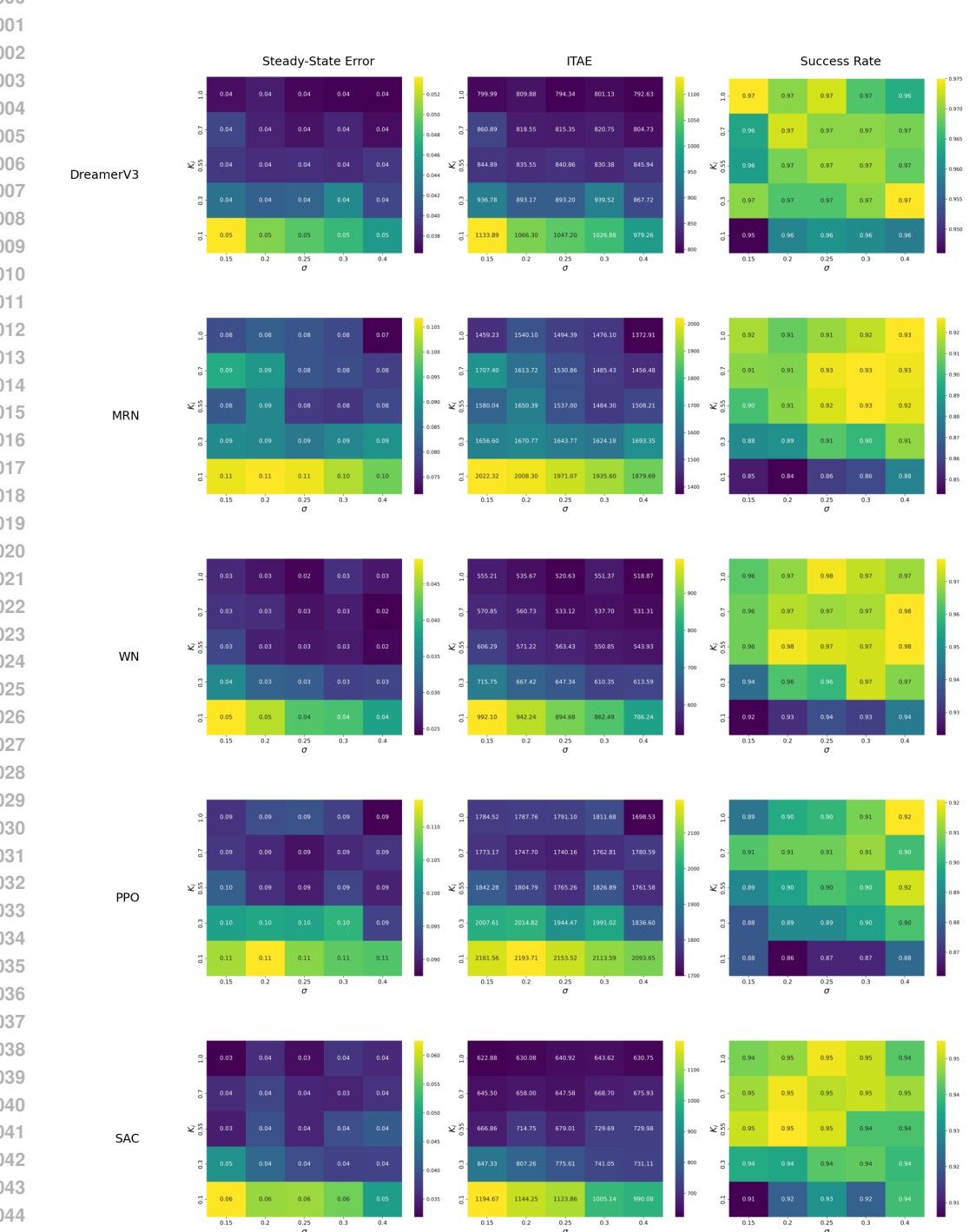

Figure 23: Combined Heatmap Visualization of Method Performance Across Metrics and Hyperparameters for EggRotate Task

## F    INTEGRAL SEPARATION ANTI-WINDUP PI CONTROL PERFORMANCE COMPARISON

To provide a comprehensive evaluation of the integral separation anti-windup PI control performance across different scenarios, we present a detailed comparison of the control responses under various conditions. Specifically, Figure 24 illustrates the nominal case ($\sigma = 0$), while Figures 25, 26, and 27 depict responses for $\sigma = 0.1$, $\sigma = 0.3$, and $\sigma = 0.6$ respectively. These visualizations collectively demonstrate the effectiveness of the baseline anti-windup PI controller with integral separation mechanism and serve as a benchmark for our proposed Adviser-Actor-Critic framework.

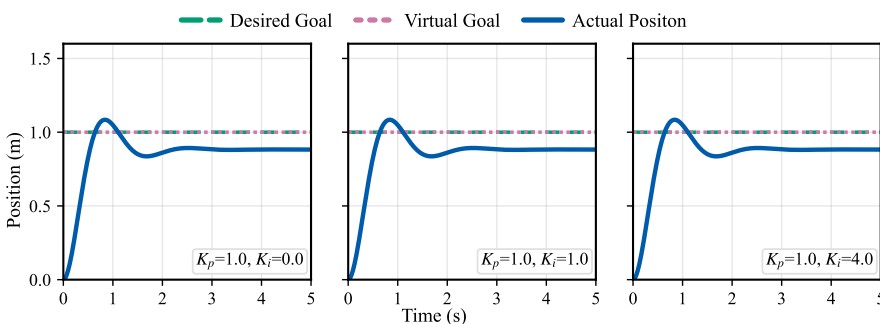

Figure 24: Anti-Windup PI Control - Case 0 (Nominal, $\sigma = 0$)

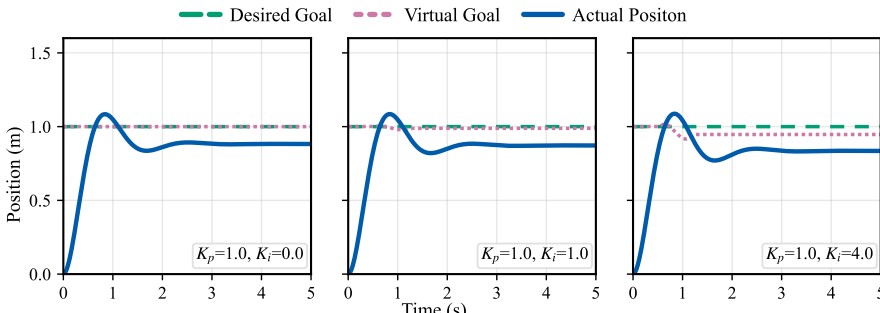

Figure 25: Anti-Windup PI Control - Case 0.1 ($\sigma = 0.1$)

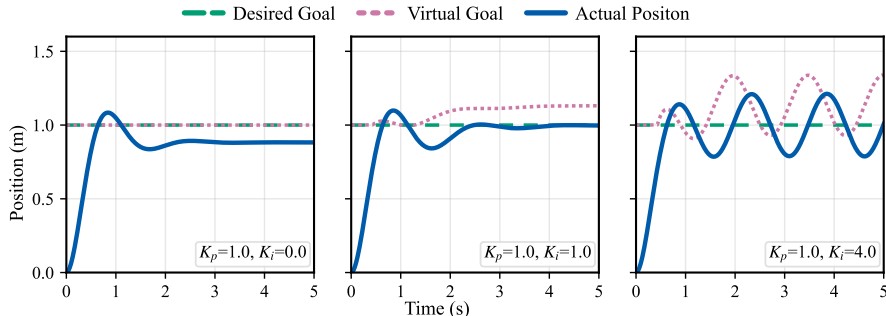

Figure 26: Anti-Windup PI Control - Case 0.3 ($\sigma = 0.3$)

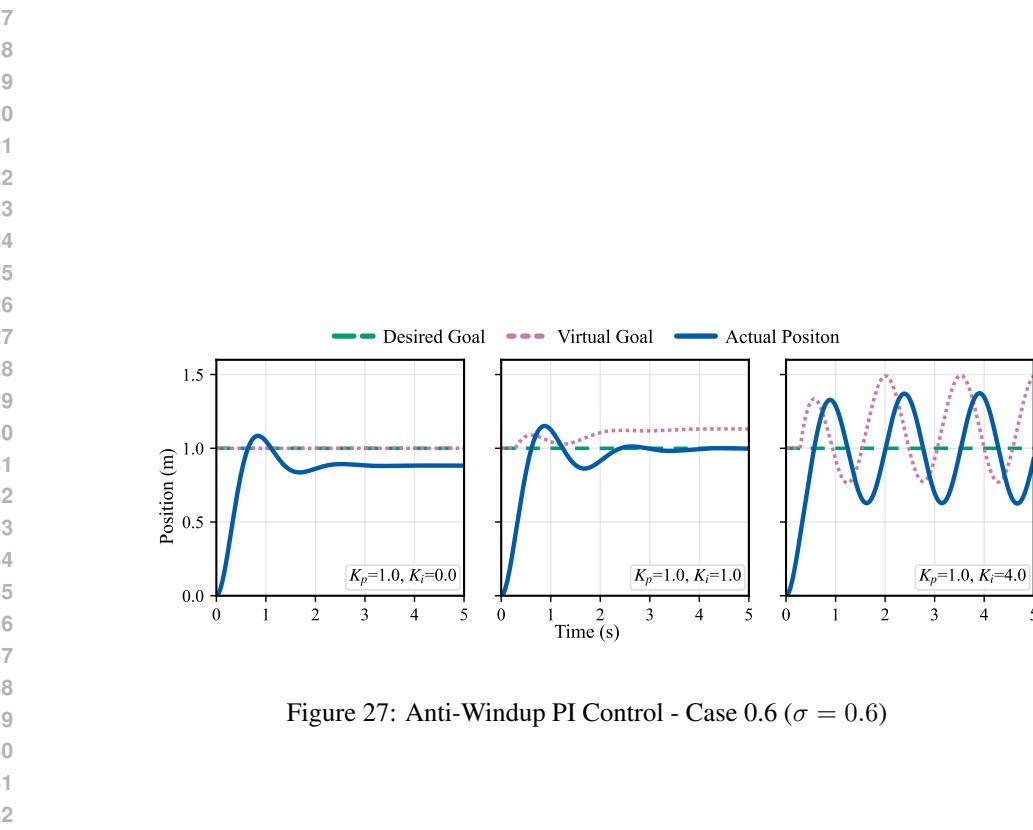

Figure 27: Anti-Windup PI Control - Case 0.6 ($\sigma = 0.6$)

