# OpenReview forum: "Adviser-Actor-Critic: Reducing Steady-State Error in Reinforcement Learning for Robotics Control"
_ICLR.cc/2026/Conference — ICLR 2026 Conference Withdrawn Submission_

### Official Review · Reviewer_RgZW · 2025-10-29

**Soundness:** 2
**Presentation:** 3
**Contribution:** 2
**Rating:** 4
**Confidence:** 4

**Summary:**

This paper proposed the Adviser-Actor-Critic (AAC) framework, targeting steady-state error reduction for high-precision robotic control tasks in reinforcement learning. AAC augments standard actor-critic architectures with an additional “adviser” module, implemented as a PI controller, that generates dynamically adjusted “virtual goals” to help the actor refine actions and reduce residual errors. The authors present a clear control-theoretic motivation, rigorous mathematical proof of zero steady-state error for constant references, and comprehensive empirical validation on both simulated (Gymnasium-Robotics benchmark tasks) and real-world (quadcopter attitude control) robotic platforms. Experimental results indicate that AAC achieves significant improvements in steady-state tracking error relative to baselines, including >80% error reduction across several benchmark tasks.

**Strengths:**

1. Strong Motivation via Real-World Challenges: The paper is motivated by the core limitation of standard RL methods: their struggle to meet sub-hardware-level precision, where even small steady-state errors matter in robotics (Section 1).

2. Novel Error Compensation Mechanism: AAC stands out by decoupling direct target acquisition from trajectory tracking through dynamic “virtual goal” adjustment, an approach that systematically counters error propagation and differs from prior goal relabeling (HER) and reward shaping paradigms (Section 4, Figure 2).

3. Integrative and Modular Architecture: The framework flexibly integrates a control-theoretic PI adviser into standard goal-conditioned RL (Figure 2), conceptualizing the adviser as a plug-in that can enable or disable compensation, which is demonstrated concisely in the degenerate case $K_i=0$ (Section 4.2).

4. Thorough Mathematical Analysis: The mathematical treatment in Appendix B, including Laplace-domain derivations and explicit transfer functions, provides a concrete, verifiable foundation. The proof regarding steady-state error elimination (Appendix B, “Theorem B.4”) is complete and well-argued under reasonable assumptions.

5. Clarity of Figures and Their Effective Use: For instance, Figure 5 effectively contrasts error compensation regimes (good vs. bad adviser tuning); Figure 7 provides qualitative insight into tradeoffs between adviser and classical PID on real hardware.

6. Honest Self-Assessment: The “Limitations” section in 6 is forthright regarding where AAC currently fails (e.g., non-steady-state settings like Slide).

**Weaknesses:**

1.Related Work Coverage: Missing Directly Relevant Literature
The paper fails to discuss several directly pertinent recent works in high-precision RL control that tackle steady-state error compensation via reward engineering, hybrid structures, or integral terms with RL. Notably, papers such as Wang et al. (2024), Han et al. (2023), Yadavari et al. (2023, 2024), and Huang et al. (2023) directly introduce error compensation mechanisms for RL-based control and should have been critically referenced and compared, particularly in Section 3 (“Related Works”) and in the empirical comparison (Sections 5.2 and D). This omission undermines the claim of distinctiveness and may inflate the paper’s originality. Judicious placement (introduction and Related Works) is essential to accurately position the proposed approach (see “Potentially Missing Related Work” for details).

2.Inadequate Coverage of Negative/Failure Cases in Main Paper
While the Slide environment is noted as a failure case, the analysis is brief and not fully explored in the main text. There is insufficient discussion of why AAC fails on environments lacking steady-state regimes or those with high stochasticity or non-stationarity, other than a passing mention in Section 5.2. The lack of deeper ablation or visualization for these negative results reduces the completeness of evidence supporting when/why AAC is (in)appropriate.

3.Potential Overfitting to Quasi-Static Tasks
The efficacy of AAC is tightly coupled with its assumption of slow-changing reference trajectories and steady-state-dominated dynamics (discussed in Section 6). There is limited empirical evaluation in more dynamic or nonstationary settings. No experiments explicitly test AAC’s robustness in the presence of fast-changing or highly stochastic targets, or with systematically degraded actors. This constrains its practical utility outside of relatively static or slowly-varying control environments.

4.Modular Adviser Design Is Not Fully Explored
While the adviser is presented as an extensible module, only the PI controller variant is evaluated. Other possible “adviser” instantiations (e.g., more sophisticated integral/observer-based, nonlinear/adaptive control, fuzzy controllers) are not empirically tested. The claim of architectural extensibility remains partially speculative.

5.Experimental Design Limitations
While Table 1 provides strong evidence for many benchmarks, the comparative baselines do not include recent hybrid RL-control or RL-with-integral-reward baselines from the missing related literature, as discussed above. Furthermore, Figure 6 and related columns focus heavily on MRN; other competitive approaches, like control-aware RL variants or hierarchical RL, are omitted. This weakens the empirical benchmarking against the state of the art.

**Potentially Missing Related Work**

Wang, L., Zheng, Z., Lin, Y. (2024): Steady-State Error Compensation for Reinforcement Learning with Quadratic Rewards — Proposes integrating integral terms into RL reward functions for steady-state error mitigation, highly relevant to the motivations and methodological space of this paper. Should be cited in Section 3 and the results tables.

Han, R., et al. (2023): Deep Reinforcement Learning Trajectory Tracking Control for a Six-Degree-of-Freedom Electro-Hydraulic Stewart Parallel Mechanism — Directly addresses high-precision trajectory tracking with deep RL, matching key application domains of this paper; needs discussion in both related work and the comparative experiments.

Yadavari, H., et al. (2023): Deep Reinforcement Learning-Based PID Controller for Position Control of Stewart Platform — Advances deep RL optimization of PID for precise control, very close in spirit to AAC’s adviser mechanism; should be added to Section 3 and result comparisons.

Huang, H.-C., et al. (2023): Reinforcement Fuzzy Q-Learning for Self-Organizing Holonomic Motion Control of Six-Link Stewart Platforms — Uses RL for precise motion control in complex robotics, directly connected to precision-centric control challenges addressed in AAC; important for positioning in Section 3.

Yadavari, H., et al. (2024): Adaptive Cruise Control Using Deep Reinforcement Learning with Integral Reward Terms — Introduces integral reward concepts for steady-state error reduction in RL, directly analogous to AAC’s objectives; should be cited and discussed in methodology and results.

Wang, L., et al. (2024): Steady-State Error Compensation in Reference Tracking and Disturbance Rejection Problems for Reinforcement Learning-Based Control — Proposes error compensation strategies for RL control, critical for Section 3 and possibly requiring experimental comparison.

Han, R., et al. (2023): Constant Force-Tracking Control Based on Deep Reinforcement Learning in Dynamic Auscultation Environment — Explores deep RL for force-tracking control, related in methodology and challenges.

Yadavari, H., et al. (2023): An Optimized Position Control via Reinforcement-Learning-Based Hybrid Structure Strategy — Hybrid RL–traditional control strategy for precision, closely aligned with the dual-loop architecture of AAC.

**Questions:**

1.Can the authors clarify why the directly relevant literature—especially those on RL with integral error compensation or hybrid/PI controllers for robotics—was omitted from Section 3? Would empirical comparison with these approaches reveal further tradeoffs or limitations?

2.Have the authors explored adviser module extensions beyond PI control (e.g., nonlinear, observer-based, or adaptive variants) in practice? What challenges were encountered? Are any performance gains or stability risks observed in preliminary trials?

3.For environments like Slide where AAC is ineffective, can the authors provide a more detailed ablation and analysis (e.g., learning curves, failure visualizations) to illuminate failure mechanisms and boundary conditions for the method’s applicability?

4. Based on the content of Figure 7, AAC only optimizes the steady-state error compared to ordinary RL, but its overshoot and settling time are worse than those of conventional PID control. So why should AAC be used in practical deployment? Why doesn't the author choose an experimental subject with more complex dynamics and nonlinearity, making RL control necessary?

---

### Official Review · Reviewer_weBt · 2025-10-29

**Soundness:** 2
**Presentation:** 1
**Contribution:** 2
**Rating:** 2
**Confidence:** 3

**Summary:**

The paper introduces Advisor-Actor-Critic (AAC), a framework that adds a classical PI controller (advisor) to a standard goal-conditioned reinforcement learning (RL) agent to reduce steady-state error (SSE) in robotic control tasks. The advisor modifies the goal given to the RL agent, creating a "virtual goal" that pushes the agent to overcompensate for and thereby eliminate residual tracking errors.

**Strengths:**

The modular advisor that can be added on top of a trained RL agent is a nice and practical idea. This enables tuning of RL policy behaviour without the need to re-train.

**Weaknesses:**

Missing details and experimental rigor:
- The authors claim that vanilla RL cannot solve high precision control tasks, which I am sceptical about. They do not provide evidence for this claim. By providing some strong state-of-the-art baseline (e.g. BRO, SimbaV2, XQC) results they could put evidence behind the claim.
- The authors do not provide crucial experimental details, e.g., how many seeds were used?
- The presentation of mean and 2 std is not a good metric in RL, as discussed by Agarwal et al. 2021. I encourage the authors to report more statistically significant metrics. Further, standard deviations are overlapping in the table for a number of runs, which makes me question the significance of the results.
- The evaluation is performaned on a very limited number of environments
- While the authors combine their method with a number of base algorithms, there are no alternative baselines (e.g., model-free RL or residual RL), making it hard to judge the overall performance of the proposed solution.

Presentation: The paper's presentation is lacking in multiple areas.
- The overall structure and writing of the paper could be improved.
- Figure 1. is not really helpful, and has no clear explanation
- I do not really understand Figure 6, and there is not really a good explaination in the text.
- Result presentation in Table 1 should be improved. Ideally the authors would provide a Figure, it's hard to identify trends and judge significance in table form with this many entries.

Related work:
- the related work and evaluation focuses on established but old RL methods. Discussion and comparison to state-of-the-art methods could be valuable.
- Alternative residual RL approaches are not discussed at all. Since these methods also combine classical control with RL, however, on an action-space level rather than a goal-space level.

References
BRO: Nauman et al. NeurIPS 2024 https://arxiv.org/pdf/2405.16158
SimbaV2: Lee et al. ICML 2025 https://arxiv.org/abs/2502.15280
XQC: Palenicek et al. arxiv 2025 https://arxiv.org/pdf/2509.25174

**Questions:**

- Could the authors provide a deeper analysis and explanation of the root causes of the claimed steady-state error in baseline RL methods?
- How many seeds did the authors use in their experiments?
- In their acknowledgment, the authors mention that LLMs assiseted with code development and algorithm implementation. Could they please clarify the extent of this assistance and specify the contribution of the LLM?

---

### Official Review · Reviewer_8MW3 · 2025-11-01

**Soundness:** 2
**Presentation:** 1
**Contribution:** 2
**Rating:** 4
**Confidence:** 4

**Summary:**

The paper proposes Adviser-Actor-Critic (AAC), a hybrid reinforcement learning and control framework that introduces an “adviser” which generates virtual goals to compensate steady-state tracking errors. The adviser is instantiated as a proportional–integral controller that proposes a virtual goal to a goal-conditioned policy. The method is evaluated in six gymnasium-robotics environments and on a real quadcopter attitude-control task, reporting sizable reductions in steady-state error. The paper also presents a theoretical argument for steady-state error elimination under several assumptions.

**Strengths:**

- The paper targets a well-motivated limitation of model-free reinforcement learning in high-precision tracking tasks: residual steady-state error and function approximation bias.
- The “virtual goal” concept is intuitive and operationally simple, making it attractive for practical deployments that need minimal changes to an existing goal-conditioned policy.
- The decoupled “plug-and-play” adviser design makes it easy to use: a trained policy can be used with or without the adviser, and the degenerate case $K\_p=1$, $K\_i=0$ recovers the baseline behavior.
- The adviser’s Proportional Integral (PI) instantiation with anti-windup is a pragmatic choice and easy to tune.
- Results are reported across multiple backbones (SAC, PPO, DreamerV3, MRN, and WN) and six gymnasium-robotics tasks, with improvements on most tasks.
- A physical quadcopter experiment demonstrates sub-degree steady-state attitude precision, and the paper carefully notes a trade-off in settling time when the adviser is enabled.

**Weaknesses:**

- The definition of the virtual goal, central to the proposed Adviser-Actor-Critic method, is inconsistent across the main text, pseudocode, and appendix. This creates ambiguity about what was actually implemented and undermines the theoretical claims built on that definition.
- The quaternion adviser update is not defined on SO(3), leading to potential drift and dimensional inconsistencies. The gating and anti-windup mechanisms for orientation control also mix units and differ between equations and pseudocode.
- Theoretical guarantees rely on strong and largely unverifiable assumptions (e.g., diagonal dominance, full-rank system matrices) that are not connected to the actual robotic implementations. This gap limits the credibility and generality of the claimed steady-state error elimination.

**Questions:**

1. Could the observed steady-state improvements be achieved with simpler goal formulations, such as sparse rewards with tolerance-based success regions or minimum-time formulation (see Vasan et al., 2024), without introducing an adviser?
2. Virtual-goal definition consistency. Can you clarify which expression of the virtual goal ($g_v$) is correct and used in practice: Eq. (6), Algorithm 1, or Appendix B? How do you reconcile the differences among these forms?
3. With $K_p = 1$ and $K_i = 0$, should $g_v = g_d$ or $g_v = g_a$? Please confirm which formulation exactly recovers the baseline policy input.
4. How do you ensure that the quaternion update remains on ( SO(3) )? Is any normalization, log–exp mapping, or projection step applied in implementation?
5. The thresholding condition compares (|v|) to a value in radians. How are these units made consistent in your implementation?
6. Is the adviser active during training, deployment, or both? If both, how do the two uses differ, and have you measured the separate contributions of each?
7. Have you compared your method against simpler variants, such as adding integral action on the action, appending integrated error to the observation, or using reward shaping, to confirm that improvements stem from goal shaping specifically?

**References**
- Vasan, G., Wang, Y., Shahriar, F., Bergstra, J., Jagersand, M., & Mahmood, A. R. (2024). Revisiting sparse rewards for goal-reaching reinforcement learning. arXiv preprint arXiv:2407.00324.

---

### Note · Authors · 2025-11-12

I have read and agree with the venue's withdrawal policy on behalf of myself and my co-authors.